# Simulating Environments for Evaluating Scarce Resource Allocation Policies

## Abstract

Consider the sequential decision problem of allocating a limited supply of resources to a pool of potential recipients: This *scarce resource allocation* problem arises in a variety of settings characterized by "hard-to-make" tradeoffs—such as assigning organs to transplant patients, or rationing ventilators in overstretched ICUs. Assisting human judgement in these choices are *dynamic allocation policies* that prescribe how to match available assets to an evolving pool of beneficiaries—such as clinical guidelines that stipulate selection criteria on the basis of recipient and organ attributes. However, while such policies have received increasing attention in recent years, a key challenge lies in *pre-deployment evaluation*: How might allocation policies behave in the real world? In particular, in addition to conventional backtesting, it is crucial that policies be evaluated on a variety of possible scenarios and sensitivities—such as distributions of recipients and organs that may diverge from historic patterns. In this work, we present `AllSim`, an open-source framework for performing data-driven simulation of scarce resource allocation policies for pre-deployment evaluation. Simulation environments are *modular* (i.e. parameterized componentwise), *learnable* (i.e. on historical data), and *customizable* (i.e. to unseen conditions), and —upon interaction with a policy—outputs a dataset of simulated outcomes for analysis and benchmarking. Compared to existing work, we believe this approach takes a step towards more methodical evaluation of scarce resource allocation policies.

# 1 Introduction

The distribution of organs for transplant is a prototypical example of the *scarce resource allocation* problem - one with salient "life-or-death" consequences that places significant pressure on decision-makers to make implicit but difficult trade-offs. To make the task more manageable, assisting human judgement in these choices are *dynamic allocation policies* that prescribe how to match each available unit of resource to a potential beneficiary. For instance, the United Network for Organ Sharing (UNOS) stipulates policies for organ allocation according to weighted organ- and patient-specific criteria, such as time on the waiting list, severity of illnesses, human leukocyte antigen matching, prognostic information, and other considerations [1–3]. Likewise, in the machine learning community, a variety of data-driven algorithms have been proposed as drop-in dynamic allocation policies, leveraging modern techniques for estimating treatment effects, predicting survival times, and accounting for organ scarcity—and often succeed in demonstrating high degrees of improvement in terms of life expectancies when deployed and evaluated on a backtested basis [4–6].

Is such demonstrated backtested performance sufficiently convincing for practitioners to adopt these developed allocation strategies? In many cases the answer is no, since there is still no standardised way in which this backtesting is actually undertaken [7]. The variety of methods that do exist share common challenges: First, when testing a target policy different from the actual policy used to generate the data, any offline evaluation method is immediately biased away from the true open-loop data-generating process [8]. Second, the evaluation methods themselves impute predicted outcomes, often with simple linear models [9, 10], which are not flexible enough to properly test more flexible machine learning methods. The compounding effect of these limitations leads to clinicians finding the results unconvincing [11], consequently limiting the use of these potentially very beneficial systems.

Historical data is also not the only aspect for which candidate policies should be compared against. In fact, it is crucial that policies be tested on a variety of scenarios and sensitivities for more robust evaluation - based on plausible possible *futures* - not just conditions that have been seen before. Real-world conditions may change after all, meaning historical data may no longer be representative of the current environment. To this end, we desire a system that allows us to test policies in such *counterfactual* scenarios for which we need *counterfactual* machine learning models [12] (cfr. Sect. 3.4 and our Appendix F)—thereby more systematically evaluating and benchmarking candidate policies before they are put into practice, as well as continuously updating existing policies based on anticipated or unanticipated changes to the environment. We propose to evaluate policies using machine learning as we illustrate in Figure 1.

As a motivating example, consider the impact of COVID-19 on the availability of organs for transplant - leading in many cases to a sudden and severe drop in supply [13]. Is it possible to reasonably measure performance of allocation policies against such a theoretical event *a priori*? Not with current data-based methods that do not contain an example of such an event. Conversely, with our simulated model we can intervene on the distribution of organs - reducing policy roll-out and testing to a simple task.

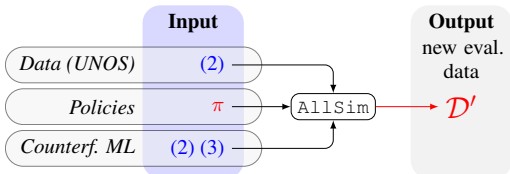

Figure 1: **Conceptual overview of `AllSim`.** We illustrate **`AllSim`**'s input and how it relates to the desiderata: (2) *learnable* and (3) *customizable*. The first desiderata (*modularity*) is discussed at length in Sect. 2 where we present **`AllSim`**'s design. In Figure 2 we discuss the (`AllSim`) node in detail. The practitioner can use `AllSim` to test a policy ($\pi$) and generate a new dataset ($\mathcal{D}'$), used to compute performance metrics of interest.

With this in mind, we wish to highlight that accurate evaluation of such allocation policies is equally, if not more, important than their development. After all, there is little point in developing policies if they cannot be shown to be beneficial and then never used. Worse, it may be that the current evaluation techniques are causing researchers to optimise for performance that may end up being *actively detrimental* in real-world deployment. And so, we now continue by considering what exactly an ideal evaluation method may consist of:

**Desiderata** We argue that a good solution should satisfy three key criteria: The environment must be (1) *modular*, in the sense that it is composed of parameterized components to allow flexible user interaction; (2) *learnable*, in the sense that it is grounded in the characteristics of real-world data; and (3) *customizable*, in the sense that it can test policies in previously unseen environment conditions. In particular, performance estimates should be unbiased when target policies different from the data-generating policies are evaluated. Note that our objective is not limited to estimating simple aggregate descriptive statistics (such as mean performance): We may also be interested in evaluating more general impacts of policy deployment—such as whether it inadvertently discriminates based on gender or ethnicity, or whether patients suffering from a particular disease are inadvertently disadvantaged.

**Contributions** In this work, we present **`AllSim`** (Allocation Simulator), a general-purpose open-source framework for performing data-driven simulation of scarce resource allocation policies for pre-deployment evaluation. We use modular environment mechanisms to capture a range of environment conditions (e.g. varying arrival rates, sudden shocks, etc.), and provide for componentwise parameters to be learned from historical data, as well as allowing users to further configure parameters for stress testing and sensitivity analysis. Potential outcomes are evaluated using unbiased causal effects methods: Upon interaction with a policy, **`AllSim`** outputs a batch dataset detailing all of the simulated outcomes, allowing users to draw their own conclusions over the effectiveness of a policy. Compared to existing work, we believe this simulation framework takes a step towards more methodical evaluation of scarce resource allocation policies.

## 2 SIMULATING ENVIRONMENTS

In this section, we explain various design choices we made when building `AllSim` in light of the three criteria we listed under "desiderata" above. Specifically, we require a method for evaluating a policy, not only for the expected outcome (for example recipient life-years), but also with respect to various demographics, aetiologies, resource waste, etc. We will solve this by outputting a synthetic dataset where recipient covariates are matched with organ covariates and an outcome. An example of an excerpt of such a dataset can be found in Appendix A.

## 2.1 PRE-DEPLOYMENT EVALUATION

Let $X \in \mathbb{R}^d$ denote the feature vector of a *recipient*, and let $\mathcal{X}(t)$ denote the arrival process of recipients. At each time $t$, let $\mathbf{X}(t) \coloneqq \{X_i\}_{i=0}^{N(t)} \sim \mathcal{X}(t)$ give the arrival set of (time-varying) size $N(t)$. Likewise, let $R \in \mathbb{R}^e$ be the feature vector of a *resource*, and let $\mathcal{R}(t)$ be the arrival process of resources. At each time $t$, let $\mathbf{R}(t) \coloneqq \{R_j\}_{j=0}^{M(t)} \sim \mathcal{R}(t)$ give the arrival set of (time-varying) size $M(t)$. While we make no assumptions how recipients are modeled, we assume that resources are immediately *perishable*—that is, each incoming resource cannot be kept idle, and must be consumed by some recipient in the same time step. In organ transplantation, for instance, the time between harvesting an organ and transplanting it ("cold ischemia time") must be minimized to prevent degradation [14–16].

Let $Y_+ \in \mathbb{R}$ be the outcome of a *matched* recipient, drawn from the distribution $\mathcal{Y}(X, R)$ induced by assigning a resource $R$ to a $X$. At each time $t$, let $\mathbf{Y}_+(t) \coloneqq \{Y_+ \sim \mathcal{Y}(X_R, R) : R \in \mathbf{R}(t)\}$ give the set of outcomes that result from matching each incoming $R \in \mathbf{R}(t)$ with its assigned $X_R$. Likewise, let $Y_- \in \mathbb{R}$ be the outcome of an *un-matched* recipient, drawn from the distribution $\mathcal{Y}(X, \varnothing)$. At each time $t$, let the set of outcomes for recipients who are never assigned a resource be given by $\mathbf{Y}_-(t) \coloneqq \{Y_- \sim \mathcal{Y}(X, \varnothing) : X \in \mathbf{X}(t), \neg(\exists t' \geq t)(R \in \mathbf{R}(t'), X = X_R)\}$. (Note that we focus on discrete-time settings (e.g. hours or days), and leave continuous time for future work). Then we have:

**Definition 1 (Scarce Resource Allocation)** Denote an *environment* with the tuple $\mathcal{E} \coloneqq (\mathcal{X}, \mathcal{R}, \mathcal{Y})$. The *scarce resource allocation* problem is to decide which recipients to assign each incoming resource to—that is, to come up with a *dynamic allocation policy* $\pi : \mathbb{R}^e \times \mathcal{P}(\mathbb{R}^d) \to \mathbb{R}^d$, perhaps to optimize some objective defined on the basis of matched/un-matched outcomes. For instance, if $Y$ is a patient's post-transplantation survival time, then an objective might be to maximize the average survival time.

**Definition 2 (Pre-Deployment Evaluation)** Let $f : \prod_k \mathbb{R}^k \times ... \to \mathbb{M}$ denote an *evaluation metric* mapping a sequence of outcomes $\{\mathbf{Y}(t)\}_{t=1,...}$ to some space of *evaluation outcome* $\mathbb{M}$ (e.g. for the average survival time, this would simply be $\mathbb{R}$), where $\mathbf{Y}(t) \coloneqq \mathbf{Y}_+(t) \cup \mathbf{Y}_-(t)$. Given a problem $\mathcal{E}$ and policy $\pi$, the *pre-deployment evaluation* problem is to compute statistics of the distribution $\mathcal{F}_{\mathcal{E}, \pi}$ of evaluation outcomes $f(\{\mathbf{Y}(t)\}_{t=1,...})$; commonly, this would be the mean $\mathbb{E}_{\mathcal{E}, \pi}[f(\{\mathbf{Y}(t)\}_{t=1,...})]$.

Note that we have defined $f$ in terms of the *sequence* of per-period outcomes such that it gives maximum flexibility: Depending on how individual outcomes $Y$ are defined, we can measure point estimates (e.g. the mean survival), compare subpopulations (e.g. whether some types of recipients systematically receive more favorable outcomes), examine trends (e.g. whether outcomes degrade as the types of recipients arriving change), or potentially investigate more complex hypotheses.

## 2.2 SIMULATION COMPONENTS

**Environment Components** Clinically validating a dynamic allocation policy not only requires estimating how that policy performs under a hypothesized scenario $\mathcal{E}$, but also how sensitive it is to changing conditions—something that is not possible using existing methods. We believe that the solution lies in *components-based simulation*. By this we mean a simulation that comprises the key components (i.e. *recipient arrival process* $\mathcal{X}$, *resource arrival process* $\mathcal{R}$, and *outcome generation mechanism* $\mathcal{Y}$), each parameterized in such way so that a practitioner can easily compose the simulation to mimic a scenario of interest. For example, if a practitioner wishes to evaluate an allocation policy against a sudden decline of resources, they need only to alter the component that is responsible for resource provision in the simulated environment. With a components-based simulation, we allow a policy to simply interact with each component. Every interaction (i.e. recipient-resource pairing) is registered in a simulated dataset. Having such a simulated dataset, the practitioner can specify typical clinical evaluation metrics to properly evaluate the policy.

**Policy Responsibilities** A `policy` has two responsibilities: (i) the `policy` maintains the set of recipients in need of a resource (being $\mathcal{P}(\mathcal{X})$), the environment only provides new recipients; and (ii) when the environment provides a new resource, the `policy` is tasked with allocating it to one of the recipients in need. We will denote these two responsibilities as `policy.add(X(t))` which allows the `policy` to maintain the arriving recipients for (i), and `policy.select(R(t))` which returns a set of recipients (one for each in $\mathbf{R}(t)$) for (ii). Note that while a policy is not part of the environment, it may naturally act differently depending on the characteristics of the environment. For instance, it can even learn how best to behave through interaction.

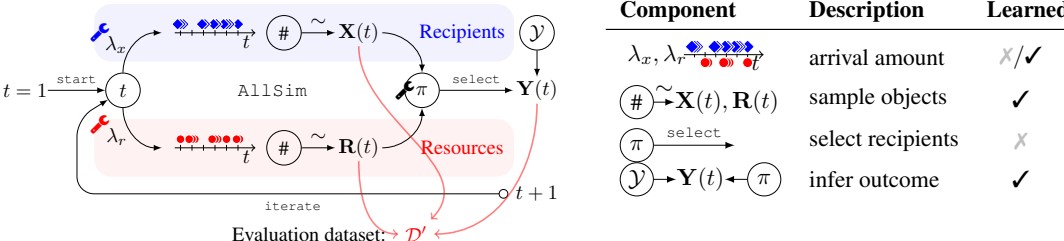

| Component | Description | Learned |
|---|---|---|
| $\lambda_x, \lambda_r$ arrival amount | arrival amount | ✗/✓ |
| # $\overset{\sim}{\rightarrow}$ $\mathbf{X}(t), \mathbf{R}(t)$ | sample objects | ✓ |
| $\pi$ select | select recipients | ✗ |
| $\mathcal{Y}$ → $\mathbf{Y}(t)$ ← $\pi$ | infer outcome | ✓ |

Figure 2: **Simulation life-cycle of** AllSim. In the leftmost part of this figure, we have illustrated the simulation life-cycle. In the rightmost part, we list each component and indicate whether or not they are learned from data. *Simulation life-cycle.* First, we sample the amount of each object arrives on day $t$. The expected amount is indicated by $\lambda_x$ and $\lambda_r$ for the recipients, and resources, respectively. The expected amount should be interpreted as an arrival rate. Once the amount of objects is specified, they are sampled from a distribution $\mathbf{X}(t)$ and $\mathbf{R}(t)$. Next, each objected is presented to the ScRAP, we wish to test, and the ScRAP replies by selecting a set of recipients whom the resources will be presented to. Finally, the simulation evaluates the ScRAP's decision, by providing an inferred outcome. Wrench icons (🔧) and an ✗'s in the table indicate which components can be user-specified, others (with ✓) are learned. Finally, $\mathbf{X}(t)$, $\mathbf{R}(t)$, and $\mathbf{Y}(t)$ compose a dataset $\mathcal{D}'$ for evaluation.

**Non-stationary Environments** A major shortcoming in simulations used to evaluate resource allocation policies in medicine, is that they cannot evaluate policies in non-stationary settings. Hence, we model the environment with a time-indicator, $\mathcal{E}(t)$. In this subsection, we discuss how we can model a non-stationary environment, while still maintaining a large level of control for the practitioner.

Having sampled a particular recipient or resource, does not influence which is sampled next. However, the probability of sampling a particular recipient, or resource, *does* change with time. For example, the ever more increasing surgical techniques allows clinicians to attempt surgery in more challenging patients, increasing the amount of patients. Similarly for resources, an organ which would not be considered for transplantation 20 years ago, may be considered today. Conversely, the recent COVID-19 pandemic is a shock which has shown an incredible drop in supply [13].

These types of environment changes are modeled through $t$. With $t$, we can model both slightly changing drifts of supply and demand (such as those that are the consequence of improving surgical techniques), as well as sudden changes (such as those caused by drastic events). We can pair a particular recipient and a resource to a probability $p(X_i|t)$ and $p(R_j|t)$ given a time $t$, and their respective arrival processes, $\mathcal{X}(t)$ and $\mathcal{R}(t)$. A major point in favour of AllSim, is the ability a practitioner has to model change. Specifically, a practitioner can learn a model of the arrival rate as a function of time and then provide that model to AllSim (as we have in Sect. 5), or the practitioner can provide a self-specified function of the arrival rate, and test the policy in an environment, alternative than the real environment.

## 3   SIMULATION LIFE-CYCLE

Given the components defined in Sect. 2.2, we are now able to formalise the high-level simulation life-cycle. In simple terms, the simulation loop will model the interaction between $\mathcal{E}$ and the policy for each consecutive day ($t = 1, 2, \dots$) until some pre-specified end ($T$). We have organised this section around Algorithm 1 where we connect each important line with a component in Figure 2 and discuss it in a separate subsection. In Figure 2, where we illustrate the interaction between each component, and clarify which parts of the simulation can be learned, and which parts should be parameterised by the practitioner— solving the second (*learnable*) and third (*costumizable*) desiderata, respectively. Having a components-based evaluation strategy solves the first desiderata from our introduction: the simulation is *modular*, i.e. each component can be reimplemented and changed at will.

### 3.1   (I) Sample recipients and resources

The first step in Algorithm 1 comprises the first and second component in Figure 2, repeated below,

$$\underbrace{\lambda_x, \lambda_r \overset{\text{arrival amount}}{\longrightarrow} t}_{\text{arrival amount}} \underbrace{\# \overset{\sim}{\rightarrow} \mathbf{X}(t), \mathbf{R}(t)}_{\text{sample objects}}.$$

Essentially, the goal in ln 4 of alg. 1 is to translate the arrival rates to a set of recipients and resources, trademarked by their features sets. With above formulation, we model the arrival processes as a

---

**Algorithm 1:** *Main simulation loop.* This simulation life-cycle acts as a section overview for Sect. 3. In our main text we discuss how each line is simulated.

**input** : Environment, $\mathcal{E}$; A resource allocation policy denoted as `pol`
**output** : Policy runtime summary

1 Start, $t = 0$;
2 **while** *simulation runs* **do**
3  $\quad t \leftarrow t + 1$ ;                                   /* iterate time */
4  $\quad \mathbf{X}(t), \mathbf{R}(t) \sim \mathcal{E}(t)$ ;   /* **(i)** sample recip.s and resources (Sect. 3.1) */
5  $\quad$ `pol.add(`$\mathbf{X}(t)$`)` ;    /* **(ii)** notify policy of recipients (Sect. 3.2) */
6  $\quad \mathbf{X}_{\mathbf{R}(t)} \leftarrow$`pol.select(`$\mathbf{R}(t)$`)` ;   /* **(iii)** allocate resources (Sect. 3.3) */
7  $\quad \mathbf{Y}_t \sim \mathcal{Y}_{\mathcal{E}}(\mathbf{X}_{\mathbf{R}(t)}, \mathbf{R}(t))$ ;             /* **(iv)** consume resources (Sect. 3.4) */
8 **end**

---

two-step approach: (1) sample the amount of objects we may expect, (2) sample the objects from a distribution. Indicated in Figure 2, only some of these component are learned, and others user-defined.

**Recipients and organs arrive randomly.** The arrival processes $\mathcal{X}(t)$ and $\mathcal{R}(t)$ are approximated as,

$$\hat{\mathcal{X}}(t, \theta_x) = \hat{\mathcal{X}}_1(t, \theta_{1,x}) \times \cdots \times \hat{\mathcal{X}}_K(t, \theta_{K,x}), \tag{1}$$

$$\hat{\mathcal{R}}(t, \theta_r) = \hat{\mathcal{R}}_1(t, \theta_{1,r}) \times \cdots \times \hat{\mathcal{R}}_L(t, \theta_{L,r}), \tag{2}$$

where $\hat{\mathcal{X}}_k$ and $\hat{\mathcal{R}}_l$ are modeled by a Poisson arrival process with arrival rate $\lambda_k(t)$ and $\lambda_l(t)$, respectively. In Equations (1) and (2) we factorise the arrival processes in $K$ and $L$ different sub-arrival processes. Each factor corresponds with some (user-defined) recipient-type (Equation (1)) and resource-type (Equation (2)). Having these factors allows us to model increasing numbers of, for example, older patients entering a transplant wait-list. Arrival rates change with $t$ and are modeled as,

$$\lambda_k(t) = \nu_k \lambda_k(0) f_k(t), \tag{3}$$

$$\lambda_l(t) = \nu_l \lambda_l(0) f_l(t), \tag{4}$$

with $\lambda_k(t), \lambda_l(t) \in \mathbb{R}_+$, and $\nu_k, \nu_l \in \mathbb{R}_+$ as a normalising constant such that the sum of all $\lambda_k$ equal some overall arrival rate $a_x$, and similarly, the sum of all $\lambda_l$ equal some overall arrival rate $a_r$. Optionally, $\nu_k$ and $\nu_l$ can be kept fixed throughout the simulation such that $a_x$ and $a_r$ vary as does $f_{k,l}(t)$, or it can be recomputed for every step $t$, such that $a_x$ and $a_r$ are kept fixed throughout the simulation. Lastly, $f_k$ and $f_l$ are continuous functions that simulate a user-specified drift. Note that these $f$ can also be a combination of multiple drift scenarios, or can be shared across different $k, l$. Having $f$, allows practitioners to very accurately describe the non-stationarity they wish to test for.

As such, we have a set of arrival rates, $\Lambda_x = [\lambda_1, ..., \lambda_K]$, with $\sum_k \lambda_k = \alpha_x$ with $\alpha_x \in \mathbb{R}_+$ as the total arrival rate of recipients. The advantage of splitting $\alpha_x$ into multiple $\lambda_k$, is that we can finetune the arrival of certain recipient types, yet allow comparison between $\alpha_x$ and $\alpha_r$ (the total arrival rate for resources). For example, the $k^{\text{th}}$ recipient type may be completely absent when a policy is launched, but over time it gradually enters the system, increasing $\alpha_x$ as a whole. Naturally, we also model the arrival of resources as we have for recipients, but left it out of discussion for clarity.

**The recipient and resource distributions.** When a recipient or a resource arrives we sample them from a distribution denoted $p_{\theta_{k,x}}(X)$ for the recipients, and $p_{\theta_{l,r}}(R)$ for the resources. These distributions are either learnt from data, or shared as an open-source (but privatised) distribution.

Let's say we have a recipient type $k$ associated with a region in $\mathcal{X}$, and an arrival rate parameter $\lambda_k \in [0, 1)$. Note that a recipient type's support may overlap with other recipient types, and that each recipient type is modeled by a distribution $p(X|k)$. In principle, we wish to keep $p(X|k)$ fixed, and learnt from data; and $\lambda_k$ (now acting as an arrival rate conditioned on the type, $k$), tunable.

### 3.2 (II) `notify policy of recipients`

After the simulation has sampled $\mathbf{X}(t)$ and $\mathbf{R}(t)$, the `policy.add(`$\mathbf{X}(t)$`)` function is called. Specifically, this notifies the policy of the arrival of new recipients, which in Figure 2 corresponds to,

$$\mathbf{X}(t), \mathbf{R}(t) \longrightarrow \boxed{\pi}.$$

Note that the above is simply a service provided by the simulation to the policy, and is thus not a learned. The reason for this is, that the organisation of in-need-recipients is entirely up to the policy. For example, more traditional policies may maintain only one priority queue [17], whereas more novel policies may maintain multiple, based on recipient and resource types [6]. As such `AllSim` remains agnostic to the tested policy, resulting in a more general-purpose framework.

### 3.3 (III) `allocate resources`

Like Sect. 3.2, step (iii) is entirely managed by the tested policy, and in Figure 2 corresponds with,

$$\pi \xrightarrow{\text{select}} \quad .$$

Selecting which recipients that get to consume the resource, is generally what differentiates policies. When a set of recipients is selected (through a `policy.select(`$\mathbf{R}(t)$`)` call in the python interface, `AllSim` retains every piece of information associated with the select-call, which is used to evaluate.

Having the simulation retain all this information, allows `AllSim` to output a dataset of past policy behaviour, much like the original (real-world) dataset we provided to the simulation in order to learn the various components. In essence, with `AllSim` we are able to sample a synthetic dataset of a counterfactual scenario such that we are able to test a policy of interest, *as if it were already in use*.

### 3.4 (IV) `consume resources`

The final component in `AllSim` is the inference-component, yielding an outcome after a resource was consumed by a recipient. Importantly, this component allows us to evaluate a policy, despite it deviating from the data used to learn the simulation. In Figure 2, this component is illustrated as,

$$\mathcal{Y} \rightarrow \mathbf{Y}(t) \leftarrow \pi \cdot$$

The outcome is a function of the resource and its recipient, making inference hard as some combinations are less observed in the original data. Essentially, the tested policy may make

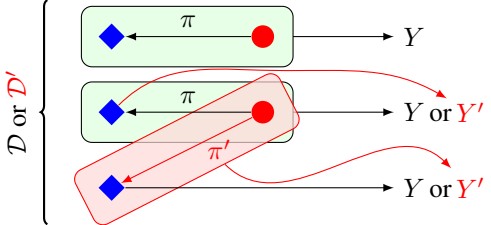

Figure 3: **Effect of a policy on data.** Above illustration, depicts two policies: $\pi$ and $\pi'$. Each policy is tasked with assigning organs (●) to recipients (◆). Besides recipients and organs, we observe an outcome, $Y$. Despite observing the same recipients and organs, a different policy results in completely different outcomes.

out-of-distribution combinations, as the simulation is learnt from data that was collected under some other policy, illustrated in Figure 3 where two policies, $\pi$ and $\pi'$ result in different datasets $\mathcal{D}$ and $\mathcal{D}'$.

**Counterfactual inference.** `AllSim` handles this by using a counterfactual estimator to infer allocation outcomes as they deal with this issue explicitly. Counterfactual methods aim to make an unbiased prediction of the potential outcome, associated with some treatment (or resource). We are interested in counterfactual methods that model the potential outcomes for the recipients when they are/are not allocated a resource. A counterfactual estimator then "completes" the simulated dataset as,

$$\mathbf{Y}(t) = \mathbb{E}[\hat{\mathbf{Y}}(\mathbf{R}(t))|\mathbf{X}_\pi(t)], \tag{5}$$

where $\hat{\mathbf{Y}}(\mathbf{R}(t))$ is the estimated potential outcome, using methodology known in the potential outcomes literature, and $\mathbf{X}_\pi(t)$ are the recipients selected by $\pi$ at time $t$. Equation (5) provides the recipient-resource pair with an estimated outcome, and presents the practitioner with a dataset:

$$\mathcal{D}^\pi := \{(X, R, \hat{Y}(R), t)_i : i = 1, \ldots, N\}.$$

$\mathcal{D}^\pi$ allows to easily calculate clinical measures of performance, as we demonstrate in Sect. 5. More information regarding these counterfactual models is provided in Appendix F.

## 4 RELATED WORK

We have summarised work related to ours in four major categories: (a) Off-policy learning and evaluation, (b) causal inference, (c) clinical simulations and trials, and (d) simulations for evaluating reinforcement learning (RL) agents. A high-level summary of these areas can be found in Table 1. For a discussion on related work in the clinical domain, we refer to our Appendix F.

Table 1: **Overview of related work.** We categorise our related work in four categories: off-policy learning, causal inference, clinical simulations, simulations for evaluating reinforcement learning agents. For each category we provide the most prominent method of calculating policy performance, and whether the categories take into account four major questions: (i) is the method tunable to new settings?; (ii) can we evaluate using data?; (iii) are the performance estimates unbiased?; and (iv) can we evaluate beyond simple aggregate descriptive statistics?

| | Citations | Perf. calculation | (i) | (ii) | (iii) | (iv) |
|---|---|---|---|---|---|---|
| Off-policy learning | [18–23] | $\frac{p_{\pi'}(R\|X)}{p_{\pi}(R\|X)}Y \sim \mathcal{D}^{\pi}$ | ✗ | ✓ | ✓ | ✗ |
| Causal inference | [24–27] | $p(R\|X)^{-1}Y \sim \mathcal{D}^{\pi}$ | ✗ | ✓ | ✓ | ✗ |
| Clinical simulations | [17, 28–31] | $\mathbb{E}_{\mathcal{D}^{\pi}}[Y\|X]$ | ✗ | ✓ | ✗ | ✓ |
| Reinforcement learning | [32–38] | $\mathbb{E}_{\text{sim}}[v(Y)\|X, R]$ | ✓ | ✗ | ✓ | ✓ |
| AllSim | (ours) | $\mathbb{E}_{\mathcal{D}^{\pi'} \sim \mathcal{X}' \times \mathcal{R}'}[Y\|X]$ | ✓ | ✓ | ✓ | ✓ |

**(a) Off-policy evaluation and (b) causal inference.** Data are collected under some active scarce resource allocation policy (ScRAP), which determines how resources are paired with recipients, which in turn determine the outcomes we get to observe. Clearly, our setting is connected to off-policy evaluation, as we wish to evaluate a policy (ScRAP) that is different from the policy that collected the available data. Figure 3 illustrates different matches (depicted as ◆──●) made across different policies resulting in different outcomes: $Y$ or $Y'$, and datasets, $\mathcal{D}$ and $\mathcal{D}'$. The difficulty of this situation, is that we observe either $\mathcal{D}$ or $\mathcal{D}'$, *not both*, relating to the potential outcomes setup [12, 39, 40]. Using $\mathcal{D}$ to evaluate $\pi$ simply means calculating some descriptive statistics. In contrast, evaluating $\pi'$ on $\mathcal{D}$ is much more involved, as the same statistics would yield a biased estimate.

One way to calculate, for example the average $Y' \sim \mathcal{D}$, is to weigh each sample: $g(◆, ●)Y \sim \mathcal{D}$. In the literature on off-policy evaluation (and learning), this is known as *importance sampling*, where each sample is weighted according to the likelihood of it belonging in $\mathcal{D}'$. Interestingly, the exact same strategy is also widely known in counterfactual learning and causal inference, under the name *inverse propensity weighting* (IPW) [20]. The key difference between both, is the target distribution. Where importance sampling transforms the estimate from one policy's distribution, to another; IPW transforms from a policy's distribution, to an unbiased estimate (i.e. to a $\frac{1}{N}$ weighting for averages).

Shown in Table 1, neither IPW nor importance sampling, provide a solution to evaluating scarce resource allocation policies. One reason for this is that the outcome is a (weighted) estimate of only one aggregate descriptive statistic (for example, expected outcome or reward). When we want additional estimates, one requires a different weighting scheme [23]. Futhermore, when we wish to test a policy in a different environment, where for example the recipients' or resources' distributions change, evaluating a policy becomes increasingly more difficult if one wishes to rely on IPW or importance sampling due to the increased differences in likelihood density. Furthermore, evaluation goes much beyond aggregate statistics, as we may be interested in, for example, demographic differences between recipients and non-recipients (Appendix A includes an example using AllSim).

**(c) Clinical evaluation and trials.** A naive method to evaluating these policies, is to model $Y'$ using simple (linear [17]) regression. Any combination ◆──●, through a new policy, $\pi'$ has an estimated outcome $\hat{Y}'$ using a regression model, trained on $\mathcal{D}$. Herein lies the problem: the regression model is still biased to $\pi$. In particular, $\mathcal{D}$ simply does not contain pairs that $\pi'$ would make, and therefor has trouble estimating $Y'$. By using causal methodology (Sect. 3.4), this is one key area where AllSim improves upon contemporary clinical simulations [9, 17, 41, 42], as these only "replay" the past without the possibility of changing the environment characteristics nor allow counterfactual inference.

Clinicians do have another way to account for this: clinical trials [28, 29]. Clinical trials estimate a causal estimand, which could then be used to "complete" the dataset as a naive regression model would above. However, two problems arise: setting up a clinical trial can sometimes be considered unethical (especially when dealing with scarce resources, and more specific research questions) [43], or clinical trials may suffer from compliance issues which will still result in biased outcomes [44, 45].

**(d) Simulations for evaluating RL agents.** A final set of solutions is that of simulations in which an RL agent can act and learn. Naturally, there are differences between policies for RL and scarce resource allocation, such as the importance of future reward which implies correlated consecutive states, and a fixed action space. However, there are also similarities, such as the definition of a policy (being a set of rules which take into account a context or state), and the online nature of the problem.

Literature on RL is blessed with some of the most well-known and actively maintained libraries for evaluating a wide range of RL algorithms [32, 33, 38]. However, other than being inapplicable in our

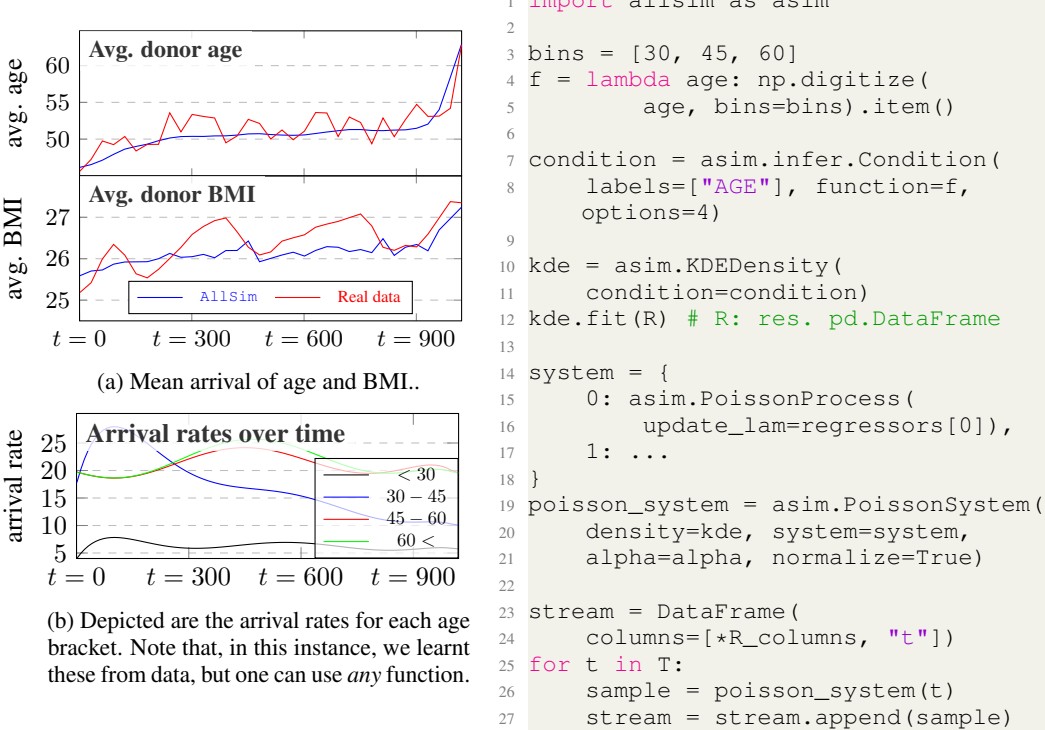

```
1  import allsim as asim
2
3  bins = [30, 45, 60]
4  f = lambda age: np.digitize(
5          age, bins=bins).item()
6
7  condition = asim.infer.Condition(
8      labels=["AGE"], function=f,
       options=4)
9
10 kde = asim.KDEDensity(
11     condition=condition)
12 kde.fit(R) # R: res. pd.DataFrame
13
14 system = {
15     0: asim.PoissonProcess(
16         update_lam=regressors[0]),
17     1: ...
18 }
19 poisson_system = asim.PoissonSystem(
20     density=kde, system=system,
21     alpha=alpha, normalize=True)
22
23 stream = DataFrame(
24     columns=[*R_columns, "t"])
25 for t in T:
26     sample = poisson_system(t)
27     stream = stream.append(sample)
```

(a) Mean arrival of age and BMI..

(b) Depicted are the arrival rates for each age bracket. Note that, in this instance, we learnt these from data, but one can use *any* function.

Figure 4: `AllSim` **can generate realistic synthetic data.** Using data on donor organs, we let `AllSim` model 3 years of organ arrivals and compare it with the actual arrival as reported in the data. In Figures 4a and 4b we show `AllSim`'s output (in donor age and BMI), given the code on the right. With minimal code, a simple condition (4 age brackets), and conservative models (polynomial regression to fit the arrival rates, and a Gaussian kernel density to model the organ densities), we find `AllSim` to accurately model the actual arrival of organs.

setting (for the reasons included above), they also have one other major downside: *they do not learn from data*. While these simulations model some very interesting environments, each aspect of the simulation is hard-coded. In order to evaluate a policy that is to be deployed in a real-world setting, we have to test it to the specifics of the environment of interest; to be defined by the practitioners developing the policy. In fact, we consider this a major reason for developing and using `AllSim`.

`AllSim` marks a significant advance over the current state-of-the-art: currently *no evaluation technique* allows a practitioner to *change the behaviour of the evaluation environment*, despite the fact that the actual environment most certainly will change in the future; nor do contemporary evaluation techniques account for potential bias from previous policies, resulting in poor performance estimates.

## 5 SIMULATION INTERFACE & EXAMPLES

Given the formal definition of `AllSim` in Sects. 2 and 3, we now introduce `AllSim`'s interface and use it directly to provide some experimental results. We split this section in two major parts: first, we discuss how we model arrival processes and resource densities (Sect. 5.1); and then, we discuss how we can combine these arrival processes in the broader simulation interface, together with a counterfactual outcome model (Sect. 5.2). Both parts provide some empirical validation of `AllSim`[1].

### 5.1 ARRIVAL PROCESSES AND RESOURCE DENSITIES

As is indicated in Figure 2 and Sect. 2.2, sampling new resources and recipients is a two-step process. On a daily basis (or some other discrete time interval in $[t]$), we first determine how many resources or recipients we need to sample, and then we sample them. The former is modeled through a Poisson arrival rate that changes over time, and the second is sampled from some learnt density. Of course, a user can implement their own arrival process by inheriting from the abstract `ArrivalProcess` class. Importantly, we need to be able to condition both on some pre-specified characteristic of the object of interest. For example, one may be interested in modeling the arrival of harvested organs of older patients distinctly from younger patients. An example of this is provided in Figure 4, where we show the changing resources coming in the system, alongside the code that generated the result.

---

[1]Note that AllSim is the first method of this kind for benchmarking scarce resource allocation policies.

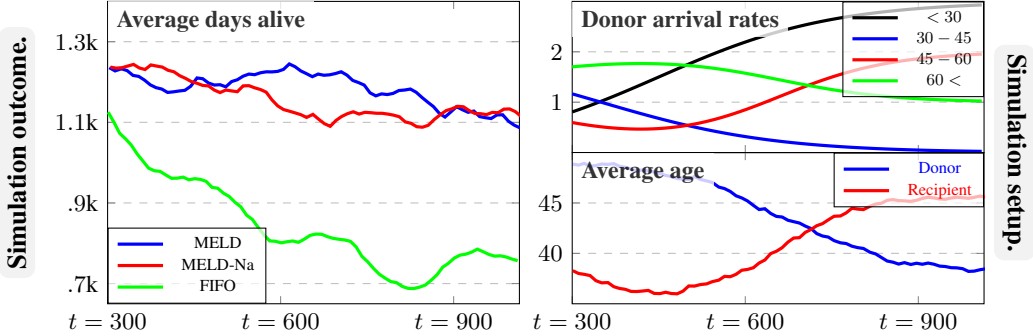

Figure 5: **Simulated outcomes.** `AllSim` can be used to compare policies in a user-specified scenario. Above we compare three different organ allocation policies: MELD, MELD-na, and FIFO. The MELD policies are clinically validated and used in practice, while FIFO is a naive first-in first-out allocation policy (which should not work at all!). When comparing these policies, our `AllSim` simulation suggests that the clinically validated policies do outperform a naive FIFO policy in terms of average survival (left), as is expected. Furthermore, the performance of each policy does seems to drop slightly over time. This too is expected as we tested the policies in a setup with increased recipient age over time (right). The reported averages are windowed over 300 samples.

**Object densities.** Before discussing a temporal arrival rate, we first discuss modeling the object's densities. Consider `lns 3-12` in the righthand side of Figure 4. Using this code, we first define what we want to condition on, using a `Condition` object: in this case we formulate age brackets. With the `KDEDensity` class, which is a subclass of the abstract `Density` class, we can automatically model a density, conditioned on these age brackets. Each `Density` object implements a `fit` and `sample` function, which is used to sample new objects by the `System`, which we discuss next.

**Arrival processes.** Using a `Density`, we move on to `lns 14-26`, where we first build a system of multiple arrival processes, one for each discrete condition as in Equations (1) and (2). In particular, we define a `PoisonProcess` for each condition (or age bracket), which is then provided to a `PoissonSystem`. Using the `PoissonSystem`, we can sample the arriving objects for each $t$ in `ln 25`. Note that we also model `alpha`, returning the overall arrival rate, such that the system can calculate an appropriate $\nu$. Figure 4a shows that `AllSim` accurately models the arriving objects.

### 5.2 COUNTERFACTUAL OUTCOMES AND SIMULATION

With the arrival processes coded above together with a counterfactual `Inference` object compose a `Simulation` object— the main interaction interface. In particular, one defines a set of arrival rates (such as in Figure 4b) for both recipients and resources (cfr. `ln 13-17`) to create a simulation:

```
simulation = asim.Sim(resource_process, patient_process, inference)
```

With that simulation, a practitioner can instantiate a `Policy`, which implements the `add` and `select` methods. For example, we have implemented the MELD policy [46], which is a widely known and used ScRAP for liver allocation. Using the `simulation`, we can generate a simulated dataset:

```
df = simulation.simulate(meld_policy, T=T)
```

Where `df` is a Pandas `DataFrame` [47, 48]. Naturally, `df` contains an enormous amount of information w.r.t. the `policy`'s allocations in our environment. As such, we have included only a subset of the potential results in Figure 5. Additional results and details can be found in Appendices A and F. Ultimately, the practitioner determines appropriate analysis, settings, and performance metrics.

## 6 CONCLUSION

`AllSim` allows standardised evaluation of resource allocation policies. While our main text heavily focuses on organ-transplantation for the sake of exhibition, Appendix E illustrates `AllSim` for COVID-19, an example outside organ-transplantation. We believe strongly that `AllSim`'s generality and modularity allows for sensible adoption in a wide range of application areas. Furthermore, having standardised evaluation will encourage research in this very important and impactful domain. Naturally, conducting research in clinical resource allocation *requires* consideration of a policy's societal impact. While we believe `AllSim` will aid in this respect also (by offering more than simple aggregate statistics), in Appendix D we provide a section on this topic specifically.

**Ethics Statement.** We envisage `AllSim` as a tool to *help* accurate and standardised evaluation of organ allocation policies, however emphasise that any finding would need to be further verified by a human expert or by a clinical trial. Ultimately, the decision on whether or not to trust a decision making tool is up to the acting clinician and hospital board. We hope that `AllSim` can help in any way to facilitate that decision, but stress that suggestions or evaluation always require critical assessment, as is the case for any research. We also refer the reader to Appendix D for a more thorough discussion on the potential societal impact of systems such as `AllSim`.

**Reproducibility Statement.** To ensure reproducibility, we have included all our code to reproduce the presented results. Furthermore, we have included a detailed discussion on how to use our simulation in Appendices B and C.

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

# A EXTENDED RESULTS

## A.1 RUNNING A SIMULATION

**A simple example.** Consider Figure 6 where we ran the MELD-Na policy in two hypothetical scenarios: one where COVID-19 happens, and one where it doesn't. With **AllSim** we can model each scenario confidently. For this particular example, we fix the seed of **AllSim** and only change the supply of organs by giving two different resource arrival processes:

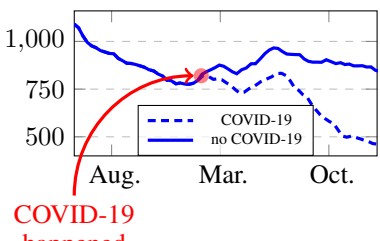

**Two hypothetical scenarios**

COVID-19 happened

Figure 6: **Two hypothetical scenarios.** We require to evaluate a policy (e.g. MELD-Na) in hypothetical (contradictory) scenarios.

```
1    def covid(t):
2    if t < 600:
3        return .5
4    else:
5        return .25
6
7    def no_covid(t):
8        return .5
```

**Tuning the simulation.** Running and composing a simulation, is as simple as defining how we wish to let the supply of resources and recipients evolve in $t$. Such evolution is expressed a changing arrival rate (crf. Equations (3) and (4). In particular, one only needs to define $\lambda_{x,r}$, normalisation is handled by AllSim. However, even there does AllSim offer detailed specification possibilities. Like $\lambda_{x,r}$, we can specify a custom function for $\alpha_{x,r}$ also. Normalisation, will then respect the value of $\alpha$ at time $t$. For example, if we wish the total arrival rate of all resources to remain constant at 5, we simply set $\alpha_r$ to a constant function:

```
1 alpha_r = lambda t: 5
```

This way, no matter how complicated our functions for $\lambda_r$, are, we know that the total arrival rate, across all conditioned distributions for the resources, we remain fixed at 5.

**Inference.** One additional component to AllSim, is its Inference module. In our simulations, we relied on OrganITE [5], where we adopted the original source-code provided by the authors, to our Inference module. Having a causal model, allows to have unbiased estimates for the resource to recipient pairing's outcome (column "Y" in Table 2).

Building a custom Inference object, is as simple as inheriting from Inference, which requires the user to implement the infer(x: np.ndarray, r: np.ndarray) -> Any function.

## A.2 ANALYSING AllSim'S OUTPUT.

Running df = simulation.simulate(policy, T=T) yields a DataFrame object, containing each match made by the policy, in the environment simulated by the ss.Simulation object. Consider Table 2 for an excerpt of the output for which we provided some aggregate results in Figure 5. Table 2 is exactly the type of data one may expect when learning about a policy, or even learning a new (ML-driven) policy [5].

Given the output of our simulation, we can see each variable, both donor as well as recipient progress over time. For example, one may be interested how each gender evolves w.r.t. allocations, with the changing supply of resources (as in Figure 5). Naturally, this is but one example of what one can achieve in terms of analysis on MELD (or any other policy), using AllSim. As such, we encourage the reader to browse through the provided code, in order to get a sense of what the possibilities really are when using AllSim.

# B DETAILS ON AllSim

While all source code is provided in our submission, we provide some additional detail on AllSim's functionality, structure, and future goals.

Table 2: **Example output.** When donor covariates are `NaN`, we know a patient died on the wait list, as they did not receive a donor organ. Note that, we only display some covariates, we have included code in our submission, should the interested reader want to inspect the complete `DataFrame`.

| Donor covariates | | Recipient covariates | | | | | |
|---|---|---|---|---|---|---|---|
| Age | BMI | INR | Sodium | Creatinine | Gender | Y | Day |
| 28.1628 | 27.7080 | 4.8144 | 138.10 | 11.351 | M | 1792.9 | 1 |
| NaN | NaN | 0.5663 | 137.71 | 0.4535 | F | 5.0 | 7 |
| NaN | NaN | 6.694 | 135.63 | 1.7323 | F | 24.948 | 26 |
| 34.893 | 21.852 | 3.8375 | 134.15 | 7.9500 | M | 257.79 | 1 |
| | | | ... | | | | |

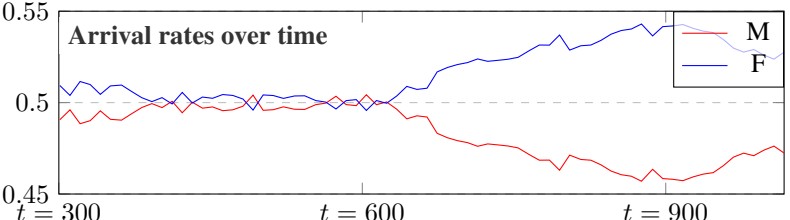

Figure 7: **Recipient-gender in function of time.** Given the MELD policy, and the changing supply of resources (cfr. Figure 5, we find that the recipient-gender changes slightly. Note that, this type of analysis is easily done given `AllSim`, as we can easily run analysis on the provided `DataFrame` object, returned by our simulation.

### B.1 NEAT FUNCTIONALITY

**One-hot encoded variables.** In heterogeneous data, one may expect a mixture of continuous and categorical variables. Good practice to handle categorical variables, is to one-hot encode them into columns of ones and zeroes, for each category. Most density estimation methods, do not automatically sample these mixtures of continuous and categorical data. Despite its performance and simplicity, a standard Kernel Density estimator, like the one we use in our simulation, cannot handle these data out-of-the-box. However, our implementation *does*. One only needs to provide the specific groups of columns which compose the categorical variables.

For example, the gender column for the recipient may be one-hot encoded into: `"GENDER_0"` and `"GENDER_1"`. When fitting our `KDEDensity`, we simply provide these columns to the `one_hot_encoded=[["GENDER_0", "GENDER_1"]]` parameter, and the `KDEDensity` automatically transforms these columns, in such a way that only one of them (being the one with maximum probability), is translated into `1`.

**Auto-scaling.** Dealing with a normalised arrival rate can be tricky to detirmine up front. As such, `AllSim` does this automatically. By solely providing an `alpha: Callable[[int], float]`, `AllSim`'s `PoissonSystem` objects will normalise any output that the `System`'s `PoissonProcess`es may have. Naturally, normalisation will depend on the chosen process, and as such does not propagate up to the general `System` nor `Process`. If, for example, on wishes to implement an alternative point process (not a Poisson process), normalisation may look differently.

**Automatic conditioning.** Another nice functionality provided in `AllSim` is automatic conditioning, in case no `Condition` is provided. A `Density` does this, by first clustering the data (from which it will later learn its density), into $K$ clusters. Naturally, one will have to provide the amount of clusters before the `Density` is able to automatically condition. Having these separate clusters, still allows defining arrival rates and processes for each cluster center, i.e. $K$ must match the amount of `Process`es are provided to the `System`.

### B.2 PACKAGE STRUCTURE

The `AllSim` package structure us as follows:

```
1  |_ src
2      |_ infer.py
3      |_ sim.py
4      |_ outcome
5          |_ counterfactual_inference.py
6          |_ counterfactual_models.py
7      |_ policies
8          |_ base.py
9          |_ policy.py
```

The two main components are in `infer.py` and `sim.py`, where `infer.py` contains everything related to `Density`, `Process`, and their subclasses; and `sim.py` concerns the `Simulation` class.

In `outcome` and `policy`, we provide basic implementations of some well known policies and counterfactual models. Note that these are actually non-essential to `AllSim`, as they can be completely user-defined, as long as they respect the `Inference` and `Policy` abstract classes, provided in the respective folders.

**Inheritance.** The base classes (`Density`, `Policy`, and `Inference`) can all be implemented by the user. Consider following class structures:

```python
class Density:
    def __init__(self
        condition: Condition, # see below for more details
        K: int=1,
        drop: np.ndarray=np.ndarray(['condition'])):
        ...

    @abstractmethod
    def sample(self, n: int=1) -> np.ndarray:
        ...

    @abstractmethod
    def fit(self, D: pd.DataFrame) -> None:
        ...

class Policy:
    def __init__(self,
        self,
        name: str,
        initial_waitlist: np.ndarray,
        dm: OrganDataModule): # a standardised datamodule should the
policy want to learn from data
        ...

    @abstractmethod:
    def select(self, resources: np.ndarray) -> Tuple[np.ndarray, np.
ndarray]:
        # returns recipients and resources, matched
        ...

    @abstractmethod
    def add(self, x: np.ndarray) -> None:
        # allows policy to add recipients to internal waitlist (if
needed)
        ...

    @abstractmethod
    def remove(self, x: np.ndarray) -> None:
        # allows policy to remove recipients, when for example they
die
        ...

class Inference
    def __init__(self,
```

```
41          model: Any,      # the Inference class acts as a wrapper for
     any model type
42          mean: float=0,   # mean and std are necessary to scale the
     outcomes, assuming
43          std: float=1):   #   standard scaling
44          ...
45
46      def __call__(self, x: np.ndarray, r: np.ndarray, *args: Any, **
     kwargs: Any) ->Any:
47          return self.infer(x, r, *args, **kwargs)
48
49      @abstractmethod
50      def infer(self, x: np.ndarray, r: np.ndarray, *args: Any, **
     kwargs: Any) -> Any:
51          ...
```

The `Condition` class then, wraps a function from a set of labels in the dataset, to a numerical value. One may choose to combine multiple variables, or just one, as we have with the `"AGE"` variable in Figures 4 and 5. Please find an example initialisation in Figure 4. While one *can* inherit from the `Condition` class, we would advice to implement a custom function (as the `lambda` function we had), and provide it to the `Condition` object.

**Future goals & open-source** Our goal is for `AllSim` to be a benchmarking standard when evaluating ScRAPs. An important milestone for this, is to completely open-source our simulation. Doing so, allows other researchers to scrutinise, enhance, and discuss how we should move beyond what we can do today. Below, we provide two points we believe our community should discuss.

## C USING `AllSim`

Here we provide the basic code, that will generate Figure 5. Naturally, the complete code is provided in the supplemental materials.

```
1    # load data
2    X, R, Y = custom_load_function(data)
3    organite = load_organite_model(location) # should load an implemented
      Inference object
4
5    # DENSITY LEARNING
6    # RESOURCES
7    bins_r = [30, 45, 60]
8    def condition_function_r(age):
9        return np.digitize(age, bins=bins_r).item()
10
11
12   condition_r = infer.Condition(
13       labels=['AGE'],
14       function=condition_function_r,
15       options=len(bins_r) + 1
16   )
17
18   kde_r = infer.KDEDensity(condition=condition_r, K=condition_r.options
     )
19   kde_r.fit(R, one_hot_encoded=groups_r)
20
21
22   # RECIPIENTS
23   bins_x = [30, 45, 60]
24
25   def condition_function_x(bilir):
26       return np.digitize(bilir, bins=bins_x).item()
27
28   condition_x = infer.Condition(
29       labels=['AGE'],
```

```
30          function=condition_function_x,
31          options=len(bins_x) + 1
32      )
33
34      kde_x = infer.KDEDensity(condition=condition_x, K=condition_x.options
        )
35      kde_x.fit(X, one_hot_encoded=groups_x)
36
37
38      # BUILD THE SYSTEMS
39      resource_system, patient_system = dict(), dict()
40
41      def update_lam_0(t):
42          return (1 / (1+np.exp(-(t-450)/150))) * 3
43
44      def update_lam_1(t):
45          return (1 / (1+np.exp((t-350)/150))) * 2
46
47      def update_lam_2(t):
48          a = (1 / (1+np.exp((t-150)/150))) * 2
49          b = (1 / (1+np.exp(-(t-650)/100))) * 2
50          return (a + b)
51
52      def update_lam_3(t):
53          a = (1 / (1+np.exp(-(t-150)/150)))
54          b = (1 / (1+np.exp((t-650)/100)))
55          return (a + b)
56
57
58      resource_system[0] = infer.PoissonProcess(update_lam=update_lam_0)
59      resource_system[1] = infer.PoissonProcess(update_lam=update_lam_1)
60      resource_system[2] = infer.PoissonProcess(update_lam=update_lam_2)
61      resource_system[3] = infer.PoissonProcess(update_lam=update_lam_3)
62
63
64      patient_system[3] = infer.PoissonProcess(update_lam=update_lam_0)
65      patient_system[2] = infer.PoissonProcess(update_lam=update_lam_1)
66      patient_system[1] = infer.PoissonProcess(update_lam=update_lam_2)
67      patient_system[0] = infer.PoissonProcess(update_lam=update_lam_3)
68
69
70      resource_process = infer.PoissonSystem(
71          density=kde_r,
72          system=resource_system,
73          alpha=lambda t: 5,
74          normalize=True)
75
76      patient_process = infer.PoissonSystem(
77          density=kde_x,
78          system=patient_system,
79          alpha=lambda t: 7,
80          normalize=True)
81
82      organite.model.eval()
83
84      simulation = sim.Sim(
85          resource_system=resource_process,
86          patient_system=patient_process,
87          inference=organite
88      )
89
90      policy = MELD(
91          name='MELD', initial_waitlist=simulation._internal_waitlist, dm=
        dm
92      )
```

```
93
94      df = simulation.simulate(policy, T=1021)
```

### C.1 GETTING STARTED

To use AllSim, a user requires at least a dataset of the following type: $\mathcal{D} \coloneqq \{(X_t, R_u, Y) : t, u \in \mathbb{N}_+\}$, with $t, u$ indicating the time of arrival. In principle, this is sufficient to start using AllSim already. In fact, this is exactly what we provided AllSim in our experiment in Figure 4. Let us elaborate:

AllSim requires to specify the following components:

1. A counterfactual model
2. The patient and resource arrival processes
3. The patient and resource densities

Components 1 and 3 are easily implemented using existing packages like econml (https://econml.azurewebsites.net) and scikit-learn (https://scikit-learn.org/stable/), respectively. They only require to use the model.fit API to be applied to the users' data:

```
1      counterf = econml.BaseCateEstimator()
2      counterf.fit(Y, R, X)
3
4
5      density_X, density_R = allsim.infer.KDEDensity(), allsim.infer.
       KDEDensity()
6      density_X.fit(X)
7      density_R.fit(R)
```

The only place where we need AllSim specifically is when we create the arrival processes, and the eventual simulation. Creating an arrival process on data, first requires us to regress the arrival probability on time:

```
1      from sklearn.preprocessing import PolynomialFeatures
2      from sklearn.linear_model import LinearRegression
3      from sklearn.pipeline import Pipeline
4
5      func_X = function(Pipeline([
6          ('poly', PolynomialFeatures(degree=6)),
7          ('linear', LinearRegression(fit_intercept=True))]))
8
9      # with amount() a function that returns how many of X arrived at t in
        D
10     func_X.fit(t, amount(X, t))
11
12     # with the arrival processes
13     patient_process = allsim.infer.PoissonProcess(func_X)
14     resource_process = allsim.infer.PoissonProcess(func_R)
15
16     patient_system = allsim.infer.PoissonSystem(density_X,
       patient_process)
17     resource_system = allsim.infer.PoissonSystem(density_R,
       resource_process)
```

The simulation is then created as:

```
1      simulation = asim.sim.Sim(
2          resource_system=resource_system,
3          patient_system=patient_system,
4          inference=counterf
5      )
```

The only thing left is to actually run the policy against our created simulation:

```
df = simulation.simulate(policy, T=100) # thats it!
```

## D   SOCIAL IMPACT

It is very clear that machine learning has the potential to transform healthcare. Its success both in other domains and already within healthcare is very promising. ML-based policies have the potential to extend lives. However, as with almost any other machine learning method, there are risks associated with their deployment in a real healthcare setting. Before any (experimental) ML-based policy (or even non-ML-based policies for that matter) are to be deployed, they requrie thourough testing.

We believe `AllSim` will enable practitioners to leverage their data and evaluate their policies in a rigorous manner. `AllSim` fully recognises the difficulties associated with evaluating policies that diverge from the data-generating policy and aims to mitigate these difficulties by employing tried and tested methods from causality. Naturally, there are some caveats: `AllSim` inherits the potential assumptions made by the counterfactual method, furthermore, if real data is used, it is crucial it is at least *somewhat* related to the environment the tested policy will end up operating in. If for example, one aims to evaluate a policy on a domain that is completely unrelated, `AllSim`'s learned simulation will provide the tested policy with unrealistic resources, recipients, and arrivals.

## E   ALLSIM IN A VACCINE DISTRIBUTION SCENARIO

AllSim is a general purpose simulator which evaluates scarce resource allocation policies. While we mainly focus on organ-transplantation in our main text, we show in this section that AllSim is also applicable in other settings. To illustrate, we show how one can implement a vaccine distribution policy evaluation system in AllSim. This use-case will show how few adjustments one has to make with respect to the presented settings in our main text.

Compared to the organ-allocation problem, in vaccine distribution, each resource is the same and they arrive in batches. Furthermore, the type of patient-in-need is also much broader (in fact, they cover the entire population). Yet, AllSim is perfectly capable of modelling this scenario given the following:

- Batch arrival simply requires a multiplier. For example, if the Poisson process samples a value of 2 on one day, we could simply interpret this as two batches of 1000 doses.
- As all vaccines are the same, we no longer require a density of resources as we required for organ allocation. This can be done by implementing a dummy-density that always returns 1 (or the amount of vaccine).
- A broader patient-type in AllSim is achieved by retraining the density of recipients over the entire population.

These implementation details are relatively simple to implement and easily done using AllSim's modular API.

While not necessarily a problem in vaccine distribution, recipient arrival in the ICU in a setting of infectious disease (such as COVID-19), is definitely different as compared to the organ-allocation setting. With organ-allocation, we can safely assume a Poisson process for recipient arrival as recipients enter the system independently. This is of course not true in an infectious disease scenario: one recipient arriving may indicate higher infection rate. As such, recipients *do not* arrive independently.

With the above, it is clear that we can no longer rely on a Poisson arrival process for recipients entering the system. Instead, to accurately model a situation of infectious disease, we recommend using a Hawkes process. To further illustrate, we include some code below showing exactly how one may go about including such a Hawkes process in AllSim.

```
class HawkesProcess(PoissonProcess):
    def __init__(
            self,
            lam: float=.1,
```

```
5            update_lam: Callable[[int], float]=lambda t: t,
6            delta: float=.1,
7            a: float=.2
8        ):
9
10        assert a >= 0, "a should be larger than or equal to 0"
11        assert delta > 0, "delta should be larger than 0"
12
13        super().__init__(lam, update_lam)
14
15        self.a, self.delta = a, delta
16        self._samples = []
17
18    def get_lam_unnormalized(self, t: int) -> float:
19        return self._baseline_lam + np.sum(
20            self.a * self.beta * np.exp(-beta * (t - self._samples[
    self._samples < t])))
21
22    def progress(self, t: int, neu: float=1) -> int:
23        self.lam = neu * self.get_lam_unnormalized(t) # eqs. (5, 6)
24        sample = np.random.poisson(lam=self.lam)
25        self._samples.append(sample)
26        return sample
```

Naturally, if recipient arrival is indeed dependent on previous arrivals, simply learning the arrival process (as we have for figure 3) should model such a self-exciting process automatically. Furthermore, the above implementation is a simple linear univariate Hawkes process. We refer to `hawkeslib` (https://github.com/canerturkmen/hawkeslib) for implementations of more types of Hawkes processes.

## F    COUNTERFACTUAL INFERENCE

`AllSim` relies for a large part on counterfactual inference. As real-world data is collected under an active policy, testing an alternative policy would almost immediately diverge from the allocations made in the data. As such, we have to *infer* an outcome from a pair made by the tested policy. If the tested policy is indeed different from the active policy, then we are unlikely to find a comparable pair in the data. The above is a question most conisdered in research on counterfactual inference [40].

Of course there are many models that perform counterfactual inference. As such, we provide a brief overview of the type of models one may resort to. Naturally, this list is non-exhaustive, but may guide a user to a model fit for their use-case.

Broadly speaking, we recognise a few "meta-categories", or *meta-learners* [49, 50]. The are as follows:

- **T-Learner.** Simplest is to learn a model (such as a neural net or random forest) for each treatment. In the vaccine distribution case that would mean learning a model on recipients who haven't received a vaccine, and a different model on recipients who have received a vaccine.
- **S-Learner.** Contrasting the above, an S-Learning learns one model, where the treatment is considered part of the covariate set. This allows for a more flexible treatment, such as continuous [51] or multivariate treatments [6]. In our main-text we use OrganITE as a counterfactual model [5], which can be considered an S-Learner.
- **X-Learner.** Used specifically for estimating the treatment effect directly, an X-Learner first learns the outcome functions (such as the T-Learner), then imputes the dataset with the completed treatmet effect (or the estimated counterfactual outcome), and then learns the treatment effect with a third model directly on the completed samples. While useful for treatment effect estimating, performing counterfactual inference with an X-Learner would default back to either an S-Learner or T-Learner.
- **R-Learner.** An R-Learner, like the X-Learner, estimates the treatment effect directly [52]. Specifically, it learns the outcome functions as well as a propensity estimate. Using these

two models, the R-Learner optimises a custom loss function based on the propensity, the outcome model, and a cross-validation setup.

- **DR-Learner.** The DR-Learner or *doubly robust*-learner is an iteration of the X-Learner [53]. Like the X-Learner the DR-learner first estimates the outcome models, then completes the dataset to learn a treatment effects model using standard supervised learning. Then the DR-Learner repeats this process using the treatment effects model from the first step.

While there are more meta-learners than what we reported above (e.g. U-Learner [52] or CW-Learner [54]), but they are much less adopted and unlike the S-Learner and T-Learner not fit for estimating the counterfactual outcomes as they, like the X-Learner, R-Learner, and DR-Learner, fit the effect function directly. We point the interested reader to the following papers: [49, 50, 52]; or to the following open-source libraries for various implementations: `causal-ml` (`https://github.com/uber/causalml`), or `econml` (`https://github.com/microsoft/EconML`).

## G    EXTENDED RELATED WORK

Let us discuss how the literature which introduces novel allocation policies, in particular to donor-organ allocation. From this, we observe that they all come up with their own unique simulation in order to validate their proposal.

**Allocation polices from medicine.** As an example, consider the following papers introduced in the medical domain (we focus here on the liver allocation setting as this is also where we focused on in our paper): [17, 55–59]. Each paper, all coming from different research groups, provide a custom simulation to validate their allocation policy. While some focus only on gathering test data (such as the recent **(author?)** [55]), others construct a simulation by simply iterating over the patients in the sequence they arrived in reality (such as **(author?)** [17]). While we would note some flaws in the way these policies are evaulated (e.g. counterfactual trajectories when allocations misalign), the truly striking observation is that *each evaluation strategy is different!* Not one paper reuses the same simulation; AllSim may change this going forward.

**Allocation policies from machine learning and OR.** The same is true for the ML [4–6, 60–64] and OR [65–69] communities. It seems that both the ML and OR community is focused more and more on this important problem– which is fantastic! But it also warrants careful evaluation. Furthermore, if we find that the evaluation strategies in medicine (which generally propose linear combinations of features [46, 70] or simple CoxPH models [17, 55, 57, 59]) have shortcomings, then this is certainly the case for much more complicated strategies introduced in ML or OR. In fact, a recent survey confirmed exactly that: [71, cfr. Limitations of ML in transplant medicine].

*Given the above, with respect to novelty, we suggset: had there been a purpose built simulator, would everyone be suggesting their own simple evaluation method paper-by-paper? We believe there is a clear literature gap which AllSim aims to fill.*

**SimUnet.** This brings us to SimUnet, which we thank the reviewer for highliging. While indeed related, SimUnet is in fact very different from AllSim. In particular, from the online README document (see `https://unos.org/wp-content/uploads/1-page-SimUnet.pdf`) it can be seen that SimUnet is more concerned to simulate *offers* from the transplant clinician's point of view; not *allocations* from an overarching healthcare system (such as UNOS, or the NHS).

