# OpenReview forum: "Simulating Environments for Evaluating Scarce Resource Allocation Policies"
_ICLR.cc/2023/Conference — Submitted to ICLR 2023_

### Official Review · Reviewer_a2x4 · 2022-10-23

**Confidence:** 3
**Correctness:** 4
**Technical Novelty And Significance:** 3
**Empirical Novelty And Significance:** 4
**Recommendation:** 8

**Clarity, Quality, Novelty And Reproducibility:**

The paper is very well written, with enough details and examples.  I am not familiar with any competing system with the same properties.

**Details Of Ethics Concerns:**

No concerns.

**Strength And Weaknesses:**

Strengths: the problem addressed is an important problem, not just for the application studies but for other potential areas of applications.  The framework proposed has multiple properties, like modularity and customizability. Modularity allows to connect it with external applications, custinmizability enables the users to decide what they need. Counterfactual inference is used to make predictions. The software is open source.
Weaknesses: there is no indication that the proposed framework has been discussed with potential users, to guarantee it provides a valuable solution for the intended users. It is still up to the practioners to decide how to configure the system, what analysis to do, etc.  If the framework has to become usable by a variety of practioners, much more guidance would be needed for specific cases. This is not difficult and does not need to be done immediately.

**Summary Of The Paper:**

The paper proposes an open-source framework to be used to allocate scarce resources to a pool of recipient.  The application example presented in the paper is allocation of organs to recipients, an application area where optimizing the allocation is extremely important.  The framework is meant to be used to explore dynamic policies for the allocation. The system used discrete time settings for the simulation, modeling arrival times for resources and for recipients and generating outcome predictions.  The framework, which learns from past data, is designed to be modular and customizable. An important aspect is the ability to do counterfactual reasoning and provide for each pair resource-recipient the potential outcome.

**Summary Of The Review:**

The paper proposes a framework for allocation of scarce resources to a pool of recipients.  The software is open source, making it easy to costumize it and extend it with additional modules.

---

> ### Author Response · Authors · 2022-11-10
> **Thank you for your review (part 1)**
>
> _Dear Reviewer a2x4, thank you very much for your positive review of our manuscript. Your comment (repeated and responded to below) is an important one as it concerns ease of adoption. We are convinced that AllSim can play an important role to validate policies; truly leveraging machine learning in practice._
>
> In your comment, we recognise three items we should respond to (emphasis ours):
>
> > there is no indication that the proposed framework [(A)] _has been discussed with potential users_, to guarantee it provides a valuable solution for the intended users. It is still up to the practitioners to decide [(B)] _how to configure the system_, what analysis to do, etc. If the framework has to become usable by a variety of practitioners, much more [(C)] _guidance would be needed for specific cases_. This is not difficult and does not need to be done immediately.
>
> - (A) Discussion with potential users
>
> - (B) How to configure the system?
>
> - (C) Guidance for specific cases
>
> Please allow us to respond to each separately.
>
> ## (A) Discussion with potential users
>
> As the reviewer points out, discussion with potential users is of vital importance. In the setting of AllSim, this mainly entails the clinical community. As such, we wish to clarify that AllSim's conception is actually _due to_ criticism from the clinical community on current practices. We have gathered these criticisms from many conversations we had with many more clinicians (both practising and researching), spread over multiple countries and continents.
>
> Naturally, due to ICLR's anonymous submission policy, we are not able to acknowledge these clinicians at this stage, but will upon acceptance. However, perhaps the reviewer is interested in published literature that acknowledges the problems AllSim aims to resolve. From Gotlieb et al. (2022) which criticises evaluation in donor-organ allocation:
>
> > [...] given the additional complexities introduced by ML algorithms, it should be ensured that models undergo rigorous but fair external validation in cohorts or simulated data.
>
> Warranting interested in specific evaluation strategies in this domain which we believe AllSim to be part of.
>
> From the same work, we also see that the clinical community is interested in seeing how environment changes impact policy effect:
>
> > For more equitable systems, studies should also consider non-clinical variables of transplantation like geographic disparity, physical compatibility between donor and recipients, and resource availability, all of which may significantly impact transplant outcomes.
>
> In particular, we address the above comment with:
>
> * __[geographic disparity.]__ As a feature of both patient and donor, we can directly interact with the density of each object directly (using `allsim.infer.Density`). By changing the density (or the conditional density through `allsim.infer.Condition`), we can immediately see the effect of different disparities influencing the tested policy. Furthermore, AllSim even allows a separate arrival process per condition through the `allsim.infer.System` API.
>
> * __[physical compatibility.]__ This is directly modelled by the counterfactual outcome model. Essentially, the outcome model is a function of exactly the compatibility between resource and recipient where compatibility is expressed in terms of reception outcome.
>
> * __[resource availability.]__ Finally, resource availability is modelled using our arrival processes, using `allsim.infer.PoissonProcess` (or any other type of arrival process, discussed above in this response).
>
> We stress that above criticism is not ours; it is _"gathered from the actual practitioners"_. AllSim is our answer to the need expressed by the clinical community.
>
> ## (B) How to configure the system?
>
> We recognise that the more specific a certain use-case becomes, the more tricky configuration will be. As such, we aim to address this using three main approaches: (i) we have provided a "getting started" guide (cfr. below), (ii) we welcome and encourage feature requests as AllSim is to be completely open-source, and (iii) with AllSim comes detailed documentation, hopefully helping users to configure their use-case.
>
> We explain (i) in this subsection of our response, and (ii) and (iii) in the next (_(C) Guidance for specific cases_)
>
> **(i) AllSim "Getting started" guide.** To use AllSim, a user requires at least a dataset of the following type: $\mathcal{D} \coloneqq \{(X_t, R_u, Y) : t, u \in \mathbb{N}_+\}$, with $t, u$ indicating the time of arrival. In principle, this is sufficient to start using AllSim already. In fact, this is exactly what we provided AllSim in our experiment in Figure 4.  Let us elaborate:
>
> AllSim requires to specify the following components:
>
> * A counterfactual model
> * The patient and resource arrival processes
> * The patient and resource densities
>
> `continued in next comment...`

---

> > ### Author Response · Authors · 2022-11-10
> > **Thank you for your review (part 2)**
> >
> > Components 1 and 3 are easily implemented using existing packages like `econml` [(https://econml.azurewebsites.net)](https://econml.azurewebsites.net) and  `scikit-learn` [(https://scikit-learn.org/stable/)](https://scikit-learn.org/stable/), respectively. They only require to use the `model.fit` API to be applied to the users' data:
> >
> > ```
> >     counterf = econml.BaseCateEstimator()
> >     counterf.fit(Y, R, X)
> >
> >
> >     density_X, density_R = allsim.infer.KDEDensity(), allsim.infer.KDEDensity()
> >     density_X.fit(X)
> >     density_R.fit(R)
> > ```
> >
> > The only place where we need AllSim specifically is when we create the arrival processes, and the eventual simulation. Creating an arrival process on data, first requires us to regress the arrival probability on time:
> >
> > ```
> >     from sklearn.preprocessing import PolynomialFeatures
> >     from sklearn.linear_model import LinearRegression
> >     from sklearn.pipeline import Pipeline
> >
> >     func_X = function(Pipeline([
> >         ('poly', PolynomialFeatures(degree=6)),
> >         ('linear', LinearRegression(fit_intercept=True))]))
> >
> >     # with amount() a function that returns how many of X arrived at t in D
> >     func_X.fit(t, amount(X, t))
> >
> >     # with the arrival processes
> >     patient_process = allsim.infer.PoissonProcess(func_X)
> >     resource_process = allsim.infer.PoissonProcess(func_R)
> >
> >     patient_system = allsim.infer.PoissonSystem(density_X, patient_process)
> >     resource_system = allsim.infer.PoissonSystem(density_R, resource_process)
> > ```
> >
> > The simulation is then created as:
> > ```
> >     simulation = asim.sim.Sim(
> >         resource_system=resource_system,
> >         patient_system=patient_system,
> >         inference=counterf
> >     )
> > ```
> >
> > The only thing left is to actually run the policy against our created simulation:
> >
> > ```
> >     df = simulation.simulate(policy, T=100) # thats it!
> > ```
> >
> > ## (C) Guidance for specific cases
> >
> > As previously discussed, AllSim welcomes (ii) feature requests, and will provide extensive and detailed (iii) documentation. We discuss these aspects below.
> >
> > **(ii) Feature requests.** AllSim is to be released as a fully open-source project. Not only will we provide all the code, but our entire development will happen in the open. An important aspect of open development is the ability for the user to provide feature requests, either through GitHub's issue tracking, or through entire pull requests. This is an important aspect of AllSim as it aims to engage ML researchers as well as the clinical community.
> >
> > Much like policy design, we believe evaluation should be developed in hybrid teams of clinicians and ML researchers; AllSim is no exception. The clinical response to AllSim (discussed above) was made possible only because we interact frequently and deeply with the clinical community. In fact, the entire issue of evaluation was raised by our clinical collaborators. We hope to continue interacting in this way by open-sourcing AllSim, perhaps even expanding the clinical response further.
> >
> > To your specific comment:
> >
> > > there is no indication that the proposed framework has been discussed with potential users, to guarantee it provides a valuable solution for the intended users.
> >
> > We believe that having this online forum attached to our repository will invite open discussion with the users of AllSim.
> >
> > **(iii) Documentation.** The above example gives a simple "getting started" example of AllSim. Of course, the true power of AllSim is the ability to change the environment to a user's preference and using counterfactual inference to provide a policy's outcome had it been applied in the specified environment. While we believe AllSim is developed in a user-friendly way, doing this is more involved that what is shown above. As such, an important element to further increase adoption as a thorough and clear documentation. This documentation will be provided as part of open-sourcing AllSim, as is common practice in many open-source projects.
> >
> > **Appendix E.** The above should provide some insight into how we aim to provide guidance. However, we have also worked out an example of other use-cases (beyond organ transplantation) in Appendix E for a vaccine distribution scenario.
> >
> > __*Overall we would like to thank the reviewer for their overwhelmingly positive review. We hope the above answers your main question. Should there still remain some comments, please, do not hesitate to ask!*__
> >
> > ### Additional references
> >
> > Neta Gotlieb, Amirhossein Azhie, Divya Sharma, Ashley Spann, Nan-Ji Suo, Jason Tran, Ani Orchanian-Cheff, Bo Wang, Anna Goldenberg, Michael Chassé, et al. The promise of machine learning applications in solid organ transplantation. NPJ digital medicine, 5(1):1–13, 2022

---

### Official Review · Reviewer_Ga5q · 2022-10-24

**Confidence:** 3
**Correctness:** 4
**Technical Novelty And Significance:** 2
**Empirical Novelty And Significance:** 3
**Recommendation:** 3

**Clarity, Quality, Novelty And Reproducibility:**


Clarity
- very clear

Quality
- the simulations and methods seem very sound

Novelty
- not really clear..

Reproducibility
- yes


**Strength And Weaknesses:**


Strengths:
- this paper & methods are very clear, and their discussion is very thorough
- the simulation module seems very sound, and they provide code
- the experiments are compelling, though only use 1-2 examples.

Weaknesses:
- it's difficult to assess novelty here. this paper synthesizes and combines many important topics related to resource allocation. but it's not clear whether this contributes any new ideas the literature.



**Summary Of The Paper:**

This paper presents a general module for data-driven simulations for resource allocation. They outline and demonstrate this codebase using an organ allocation scenario.


**Summary Of The Review:**

This paper is interesting and relevant and well written. The simulation module does seem very useful, and the experiments are compelling. However I'm not sure how to assess novelty here. While the module appears to be very useful, it's not clear that it contributes anything new to the literature. (Aside -- ICLR seems like a strange venue for this paper.) Nevertheless, this seems to me like a strong contribution.

If the authors can clarify the novelty of this paper (perhaps using additional simulations, discussion, applications, etc.), I would advocate strongly to accept this paper.


---

Updated this review after reading comments and other reviews. See my comment from Nov. 18.

---

> ### Author Response · Authors · 2022-11-10
> **Thank you for your review (part 1)**
>
> _Dear Reviewer Ga5q, thank you very much for your positive review of our manuscript. As your only comment of our paper is one of novelty, we focus our entire rebuttal on providing evidence that our simulation is necessary and novel._
>
> **Evaluating _without_ AllSim -- Necessity.** As an example, consider the papers introduced in the medical domain (we focus here on the liver allocation setting as this is also where we focused on in our paper):  (Kim et al., 2021; Nitski et al., 2021; Kartoun, 2022; Jochmans et al., 2017; Goldberg et al., 2021; Neuberger et al., 2008). Each paper, all coming from different research groups, provide a custom simulation to validate their allocation policy. While some focus only on gathering test data (such as the recent Kim et al. (2021)), others construct a simulation by simply iterating over the patients in the sequence they arrived in reality (such as Neuberger et al. (2008)). While we recognise some flaws in the way these policies are evaluated (e.g. counterfactual trajectories when allocations misalign), the truly striking observation is that _each evaluation strategy is different!_ Not one paper reuses the same simulation.
>
> The same is true for the ML (Berrevoets et al., 2020, 2021; Yoon et al., 2017; Connor et al., 2021; Medved et al., 2018a; Dorado-Moreno et al., 2017; Medved et al., 2018b; Yoo et al., 2017) and OR (Bertsimas et al., 2013, 2012; Elalouf et al., 2018; Dickerson et al., 2013; Akan et al., 2012) communities. It seems that both the ML and OR community is focused more and more on this important problem-- which is fantastic! But it also warrants for careful evaluation. Furthermore, if we find that the evaluation strategies in medicine (which generally propose linear combinations of features  (Kamath et al., 2001; Ruf et al., 2005) or simple CoxPH models (Kartoun, 2022; Neuberger et al., 2008; Goldberg et al., 2021; Kim et al., 2021)) have shortcomings, then this is certainly the case for much more complicated strategies introduced in ML or OR. In fact, a recent survey confirmed exactly that: : (Gotlieb et al., 2022, cfr. Limitations of ML in transplant medicine).
>
> _Given the above, with respect to novelty, we suggest: had there been a purpose built simulator, would everyone be suggesting their own simple evaluation method paper-by-paper? We believe there is a clear literature gap which AllSim aims to fill._
>
> **There are no good alternatives -- Novelty.** An obvious argument is to look at potentially similar simulators. While the above clearly demonstrates that, even {\it if} they exist, potential simulators are simply not used. Why is this? One potential candidate, specific to the liver allocation problem is LivSim  (Kilambi et al., 2018) (which we compare against already in our manuscript), a module in the Liver Simulated Allocation Module ([LSAM](https://www.srtr.org/requesting-srtr-data/simulated-allocation-models/)). LivSim has some major issues which we resolve with AllSim:
>
> **[Counterfactual trajectories]** LivSim estimates outcomes using a linear CoxPH model. Using these Cox models, they aim to provide an estimated trajectory for deviating policies. Of course, it is long known in the treatment effects literature (which we base ourselves on) that simple inference is not sufficient in these counterfactual scenarios. AllSim recognises this and performs inference using treatment effects models.
>
> **[Prespecified]** LivSim (and LSAM) are unable to learn from data. Their internal outcome module is pretrained and the model's weights are provided with and part of the actual simulation module. Doing this does not allow LivSim to be used beyond liver allocation. Even more so, LivSim can only be used in the same area where the LivSim model was trained on (USA).
>
> **[Not modular]** Contrary to AllSim, LivSim does not allow to test a policy against perturbation in the environment where the policy should operate. This is a major shortcoming! We strongly believe that a policy should be robust against all sorts of external shocks.
>
> Above concerns are not ours alone, in fact the LSAM allocation procedure has come under recent scrutiny by the clinical community (Wood et al., 2021), further motivating the research gap we aim to fill with AllSim.
>
>
> `continued in next comment...`

---

> > ### Author Response · Authors · 2022-11-10
> > **Thank you for your review (part 2)**
> >
> > **Open-source -- Clinical collaboration.** Our simulator will be entirely open-sourced. Not only would that mean that our simulator will remain entirely free for use (and always will be), it also functions as a platform for collaboration between clinicians and machine learning researchers: a clinician could request features (or even provide them directly through pull requests), which in turn would encourage machine learning researchers to handle this additional difficulty. While LivSim (discussed above) is open-sourced, its last update stems from 4 years ago. As such, _there is currently no such platform available!_
> >
> > _We would like to thank the reviewer once more for their time and effort put into their review. We truly appreciate you understanding the importance of this work and consequentially your advocacy for recommending to accept our paper. Should there still remain questions/comments, please, do not hesitate to ask!_
> >
> > ### Additional references
> >
> > W Ray Kim, Ajitha Mannalithara, Julie K Heimbach, Patrick S Kamath, Sumeet K Asrani, Scott W Biggins, Nicholas L Wood, Sommer E Gentry, and Allison J Kwong. Meld 3.0: the model for end-stage liver disease updated for the modern era. Gastroenterology, 161(6):1887–1895, 2021
> >
> > Osvald Nitski, Amirhossein Azhie, Fakhar Ali Qazi-Arisar, Xueqi Wang, Shihao Ma, Leslie Lilly, Kymberly D Watt, Josh Levitsky, Sumeet K Asrani, Douglas S Lee, et al. Long-term mortality risk stratification of liver transplant recipients: real-time application of deep learning algorithms on longitudinal data. The Lancet Digital Health, 3(5):e295–e305, 2021
> >
> > Uri Kartoun. Towards optimally replacing the current version of meld. Journal of Hepatology, 2022 Ina Jochmans, Marieke van Rosmalen, Jacques Pirenne, and Undine Samuel. Adult liver allocation in eurotransplant. Transplantation, 101(7):1542–1550, 2017
> >
> > David Goldberg, Alejandro Mantero, Craig Newcomb, Cindy Delgado, Kimberly Forde, David Kaplan, Binu John, Nadine Nuchovich, Barbara Dominguez, Ezekiel Emanuel, et al. Development and validation of a model to predict long-term survival after liver transplantation. Liver transplantation, 27(6):797–807, 2021
> >
> > James Neuberger, Alex Gimson, Mervyn Davies, Murat Akyol, John O’Grady, Andrew Burroughs, MarkHudson, UK Blood, et al. Selection of patients for liver transplantation and allocation of donated livers in the uk. Gut, 57(2):252–257, 2008
> >
> > Jeroen Berrevoets, James Jordon, Ioana Bica, Alexander Gimson, and Mihaela van der Schaar. OrganITE: Optimal transplant donor organ offering using an individual treatment effect. In Advances in Neural Information Processing Systems, volume 33, pp. 20037–20050. Curran Associates, Inc., 2020
> >
> > Jeroen Berrevoets, Ahmed Alaa, Zhaozhi Qian, James Jordon, Alexander ES Gimson, and Mihaela Van Der Schaar. Learning queueing policies for organ transplantation allocation using interpretable counterfactual survival analysis. In International Conference on Machine Learning, pp. 792–802. PMLR, 2021
> >
> > Jinsung Yoon, Ahmed Alaa, Martin Cadeiras, and Mihaela Van Der Schaar. Personalized donor-recipient matching for organ transplantation. In Proceedings of the AAAI Conference on Artificial Intelligence, volume 31, 2017

---

> > > ### Author Response · Authors · 2022-11-10
> > > **References continued**
> > >
> > > Katie L Connor, Eoin D O’Sullivan, Lorna P Marson, Stephen J Wigmore, and Ewen M Harrison. The future role of machine learning in clinical transplantation. Transplantation, 105(4):723–735, 2021
> > >
> > > Dennis Medved, Pierre Nugues, and Johan Nilsson. Simulating the outcome of heart allocation policies using deep neural networks. In 2018 40th Annual International Conference of the IEEE Engineering in Medicine and Biology Society (EMBC), pp. 6141–6144. IEEE, 2018a
> > >
> > > Manuel Dorado-Moreno, María Pérez-Ortiz, Pedro A Gutiérrez, Rubén Ciria, Javier Briceño, and César Hervás-Martínez. Dynamically weighted evolutionary ordinal neural network for solving an imbalanced liver transplantation problem. Artificial Intelligence in Medicine, 77:1–11, 2017
> > >
> > > Dennis Medved, Mattias Ohlsson, Peter Höglund, Bodil Andersson, Pierre Nugues, and Johan Nilsson. Improving prediction of heart transplantation outcome using deep learning techniques. Scientific reports, 8 (1):1–9, 2018b
> > >
> > > Kyung Don Yoo, Junhyug Noh, Hajeong Lee, Dong Ki Kim, Chun Soo Lim, Young Hoon Kim, Jung Pyo Lee, Gunhee Kim, and Yon Su Kim. A machine learning approach using survival statistics to predict graft survival in kidney transplant recipients: a multicenter cohort study. Scientific reports, 7(1):1–12, 2017
> > >
> > > Dimitris Bertsimas, Vivek F Farias, and Nikolaos Trichakis. Fairness, efficiency, and flexibility in organ allocation for kidney transplantation. Operations Research, 61(1):73–87, 2013
> > >
> > > Dimitris Bertsimas, Vivek F Farias, and Nikolaos Trichakis. On the efficiency-fairness trade-off. Management Science, 58(12):2234–2250, 2012
> > >
> > > Amir Elalouf, Yael Perlman, and Uri Yechiali. A double-ended queueing model for dynamic allocation of live organs based on a best-fit criterion. Applied Mathematical Modelling, 60:179–191, 2018
> > >
> > > John P Dickerson, Ariel D Procaccia, and Tuomas Sandholm. Failure-aware kidney exchange. In Proceedings of the fourteenth ACM conference on Electronic commerce, pp. 323–340, 2013
> > >
> > > Mustafa Akan, Oguzhan Alagoz, Baris Ata, Fatih Safa Erenay, and Adnan Said. A broader view of designing the liver allocation system. Operations research, 60(4):757–770, 2012
> > >
> > > Patrick S Kamath, Russell H Wiesner, Michael Malinchoc, Walter Kremers, Terry M Therneau, Catherine L Kosberg, Gennaro D’Amico, E Rolland Dickson, and W Ray Kim. A model to predict survival in patients with end-stage liver disease. Hepatology, 33(2):464–470, 2001
> > >
> > > Andres E Ruf, Walter K Kremers, Lila L Chavez, Valeria I Descalzi, Luis G Podesta, and Federico G Villamil. Addition of serum sodium into the meld score predicts waiting list mortality better than meld alone. Liver Transplantation, 11(3):336–343, 2005
> > >
> > > Neta Gotlieb, Amirhossein Azhie, Divya Sharma, Ashley Spann, Nan-Ji Suo, Jason Tran, Ani Orchanian-Cheff, Bo Wang, Anna Goldenberg, Michael Chassé, et al. The promise of machine learning applications in solid organ transplantation. NPJ digital medicine, 5(1):1–13, 2022
> > >
> > > Vikram Kilambi, Kevin Bui, and Sanjay Mehrotra. Livsim: an open-source simulation software platform for community research and development for liver allocation policies. Transplantation, 102(2), 2018
> > >
> > > Nicholas L Wood, Douglas B Mogul, Emily R Perito, Douglas VanDerwerken, George V Mazariegos, Evelyn K Hsu, Dorry L Segev, and Sommer E Gentry. Liver simulated allocation model does not effectively predict organ offer decisions for pediatric liver transplant candidates. American Journal of Transplantation, 21(9):3157–3162, 2021

---

> ### Comment · Reviewer_Ga5q · 2022-11-18
> **Update following author comments**
>
> Thank you for your responses to my questions.
>
> After reading through these comments, and the other reviews, I decided to lower my recommendation to 3 "reject".
>
> My reasoning is:
>
> This paper does not go far enough to demonstrate the importance of a common simulation framework. The authors make a compelling argument that a standardized simulation environment for resource allocation is needed, and they discuss the shortcomings of other published simulations. I fully agree with their argument. They also present a simulation environment + codebase, and they demonstrate that it works (also great).
>
> But they do not demonstrate the novelty or relevance of their methods. As other reviewers point out, many other researchers have created simulation environments for resource allocation and released their code publicly. The authors of this paper claim that their environment is better for several reasons (Section 4, and specifically Table 1). However they do not demonstrate that their environment is better in any meaningful way. Below I provide some suggestions for how to do this. Other reviewers also had good suggestions.
>
> Overall: I think that this work is very important, and I intend for my feedback to be constructive. I hope that the authors continue to refine this paper and I hope to see an updated version in the future.
>
> Suggestions:
>
> 1) Technical comparison. To show that your environment is better than the "simple" environments used in previous studies, you can compare simulation results from your environment to those of previous work. (You should probably implement their simple environments in order to have a fair comparison.) I would hope that in some cases your environment will produce different results than theirs. Since your environment is more flexible, you can also investigate why you find different results. This insight is crucial for practitioners who use simulation results to guide decision-making. Use this as evidence that your environment is truly better than simpler methods, and you can present this as a use-case so that readers can imagine how to use your methods in their work.
>
> 2) User study. You can also demonstrate the value of your methods by engaging directly with practitioners in relevant resource-allocation settings, like organ allocation. Maybe these practitioners will have a novel suggestion for how to use your simulation environment, or maybe they can provide feedback about what makes a "good" or "bad" simulation in their view. You can do this through a small interview study or survey, or even by including a practitioner as a co-author on your paper. This would make your contributions more meaningful in my view.

---

> > ### Author Response · Authors · 2022-11-19
> > **Please reconsider**
> >
> > Dear Reviewer Ga5q,
> >
> > Thank you for taking the time to respond to our rebuttal, this is much appreciated.
> >
> > We are incredibly disappointed by your score decrease. Especially, since your review (and your response also) seem very positive. Even so much as expressing your strong advocacy to accept our paper!
> >
> > Respectfully, we believe your score decrease to be misguided. Please allow us to explain, in hopes to convince you to reconsider.
> >
> >
> >
> > **1: Technical comparison.** At first glance it would seem obvious to ask for comparisons of currently in use systems, however, it is actually not informative. As an example: one of these practices can achieve a simulation that _perfectly_ models reality. It just replays what is observed in the data. A comparison with this trivial simulation would render any other benchmark as lesser performant, but this is not the case: This trivial simulation assumes we are testing the same policy as the active policy when the data was collected... (which is never the case)
> >
> > The reason we cannot compare with AllSim on a technical level, is because AllSim is the first of its kind. AllSim allows _counterfactual_ scenarios, a functionality that **no other simulation/evaluation strategy can offer**. In fact, this is one of the reasons we believe AllSim to be an important contribution.
> >
> >
> >
> > **2: User study.** We want to clarify: AllSim is _conceived by_, _developed under supervision by_, and _evaluated by_ a large group of respected clinicians (both practicing and researching) that are active in the field, spread over multiple hospitals, countries, and continents. AllSim is not a simulation "cooked up" by people that are unaware of clinical practice. We will acknowledge this group of clinicians, but cannot at this stage due to ICLR's double blind submission policy.
> >
> > However, we _can_ refer to work in the public domain, essentially motivating AllSim's conception.
> >
> > From the recent Goetlieb et al. (2022):
> >
> > > For more equitable systems [allocation policies], studies should also consider non-clinical variables of transplantation like geographic disparity, physical compatibility between donor and recipients, and resource availability, all of which may significantly impact transplant outcomes.
> >
> > As they are currently not considered, AllSim offers a solution to _all_ of the problems raised (by the clinical community) above.
> >
> > * __[geographic disparity.]__ As a feature of both patient and donor, we can directly interact with the density of each object directly (using `allsim.infer.Density`). By changing the density (or the conditional density through `allsim.infer.Condition`), we can immediately see the effect of different disparities influencing the tested policy. Furthermore, AllSim even allows a separate arrival process per condition through the `allsim.infer.System` API.
> >
> > * __[physical compatibility.]__ This is directly modelled by the counterfactual outcome model. Essentially, the outcome model is a function of exactly the compatibility between resource and recipient where compatibility is expressed in terms of reception outcome.
> >
> > * __[resource availability.]__ Finally, resource availability is modelled using our arrival processes, using `allsim.infer.PoissonProcess}` (or any other type of arrival process, discussed above in this response).
> >
> > We stress that above criticism is not ours; it is _gathered from the actual practitioners_. AllSim is our answer to the need expressed by the clinical community.
> >
> >
> >
> >
> > **_With the above, we kindly ask the reviewer to reconsider their stance. In your review and later response, you pin-point exactly why AllSim is an important contribution. Through AllSim the clinical community would benefit greatly from the know-how in the machine learning community._**

---

### Official Review · Reviewer_uUjW · 2022-10-25

**Confidence:** 3
**Correctness:** 3
**Technical Novelty And Significance:** 3
**Empirical Novelty And Significance:** 2
**Recommendation:** 5

**Clarity, Quality, Novelty And Reproducibility:**

The paper is generally written well and easy to follow. The motivations, key components of the simulator and their usage are explained clearly. The authors shared the source code of the simulator, so it can be easily reproduced. Related works are exhaustive and covered relevant papers in different settings. My only concern remains with the technical novelties and practical deployment of the simulator in the real-world as mentioned earlier.

**Strength And Weaknesses:**

Scarce resource allocation (e.g., Kidney/Liver matching) is an extremely important and challenging problem that requires online planning. While there has been significant effort in AI and OR communities to find optimal online matching of demand and supplies of these scarce resources, we absolutely need an accurate simulation environment for evaluation of different decisions. The proposed simulator can be used for counterfactual evaluation of policies learned by offline RL methods from a batch dataset. The simulator provides end-to-end functionalities from generating demand and supply of resources, allocating resources and evaluating a policy by supporting counterfactual inference. Having said that, I have some reservations regarding the technical novelties of the paper. There have been efforts for generating simulation environments for kidney or liver matching or transplantation (e.g., SimUNet: https://unos.org/solutions/simunet/). Does the simulator generalize to different settings of organ matching with simple tunable knobs? Moreover, tuning these knobs could be challenging for the practitioners as the outcome might depend on a variety of indicators for individual patients. I think it would be great if the authors can perform a human evaluation study and report statistics in terms of how much value the actual practitioners see in this simulator. If it does not pass the bar of practitioner’s expectation, then the counterfactual evaluation in the simulator might not be of practical use. On the other hand, I was expecting some guidance on how to tune the knobs of the simulator, especially for accelerating research progress in this domain.

**Summary Of The Paper:**

The paper proposed a simulation environment for evaluating scarce resource (e.g., Kidney, Liver) allocation policy. The proposed allocation simulator (AllSim) is modular, learnable and customizable. The simulator provides functionality for generating resource and recipient distribution, allocating resources, and evaluating performance of the allocation. It allows practitioners to properly tune different knobs of the simulator to match their practical needs. The paper also demonstrates the performance of different components experimentally.

**Summary Of The Review:**

The paper proposed a novel simulator with end-to-end functionalities for evaluating a scarce resource allocation policy, especially in medical domain setting. An accurate simulator would be hugely effective for research progress in the medical domain as well as offline RL space. I appreciate the specific functionalities of resource allocation and counterfactual inference, and their detailed explanation in experimental results. Moreover, the simulator could be easily reproduced as the source codes are shared. Having said that, I still have some reservations regarding the technical/methodological contributions, e.g., how well it generalizes to different organ settings and how much finetuning is required are not clear to me, and its practical adoption, e.g., I expected an human evaluation on how satisfied the practitioners are with the simulator, or how much effort the practitioners need to finetune all the knobs, or even if it is feasible to properly tune all the knobs to generate accurate behavior or outcomes of a patient health?

---

> ### Author Response · Authors · 2022-11-10
> **Thank you for your review (part 1)**
>
> _Thank you for your thoughtful comments and suggestions! Per the the points raised in your review, we give answers to each in turn, as well as pointing out corresponding updates to the revised manuscript. We will split our response into 3 for added clarity:_
>
> - (A) Novelty and SimUnet
>
> - (B) Generalisation of AllSim
>
> - (C) Simulation parameters and adoption
>
> ## (A) Novelty and SimUnet
>
> Concerning your comment:
>
> > I have some reservations regarding the technical novelties of the paper. There have been efforts for generating simulation environments for kidney or liver matching or transplantation (e.g., SimUNet: [https://unos.org/solutions/simunet/](https://unos.org/solutions/simunet/)).
>
> Before comparing AllSim to SimUnet, let us first reiterate that the whole literature introducing novel allocation policies all come up with their own unique simulation in order to validate their proposal.
>
> **Allocation policies from medicine.** As an example, consider the following papers introduced in the medical domain (we focus here on the liver allocation setting as this is also where we focused on in our paper):  (Kim et al., 2021; Nitski et al., 2021; Kartoun, 2022; Jochmans et al., 2017; Goldberg et al., 2021; Neuberger et al., 2008). Each paper, all coming from different research groups, provide a custom simulation to validate their allocation policy. While some focus only on gathering test data (such as the recent Kim et al. (2021)), others construct a simulation by simply iterating over the patients in the sequence they arrived in reality (such as  Neuberger et al. (2008)). While we would note some flaws in the way these policies are evaluated (e.g. counterfactual trajectories when allocations misalign), the truly striking observation is that _each evaluation strategy is different!_ Not one paper reuses the same simulation; AllSim has the potential to change this going forward.
>
> **Allocation policies from machine learning and OR.** The same is true for the ML (Berrevoets et al., 2020, 2021; Yoon et al., 2017; Connor et al., 2021; Medved et al., 2018a; Dorado-Moreno et al., 2017; Medved et al., 2018b; Yoo et al., 2017) and OR (Bertsimas et al., 2013, 2012; Elalouf et al., 2018; Dickerson et al., 2013; Akan et al., 2012) communities. It seems that both the ML and OR community is focused more and more on this important problem-- which is fantastic! But it also warrants careful evaluation. Furthermore, if we find that the evaluation strategies in medicine (which generally propose linear combinations of features (Kamath et al., 2001; Ruf et al., 2005)) or simple CoxPH models (Kartoun, 2022; Neuberger et al., 2008; Goldberg et al., 2021; Kim et al., 2021)) have shortcomings, then this is certainly the case for much more complicated strategies introduced in ML or OR. In fact, a recent survey confirmed exactly that:  (Gotlieb et al., 2022, cfr. Limitations of ML in transplant medicine).
>
> _Given the above, with respect to novelty, we suggest: had there been a purpose built simulator, would everyone be suggesting their own simple evaluation method paper-by-paper? We believe there is a clear literature gap which AllSim aims to fill._
>
>
> **SimUnet.** This brings us to SimUnet, which we thank the reviewer for highlighting. While indeed related, SimUnet is in fact very different from AllSim. In particular, from the online README document (see [https://unos.org/wp-content/uploads/1-page-SimUnet.pdf](https://unos.org/wp-content/uploads/1-page-SimUnet.pdf)) it can be seen that SimUnet is more concerned to simulate _offers_ from the transplant clinician's point of view; not _allocations_ from an overarching healthcare system (such as UNOS, or the NHS).
>
> From SimUnet's document:
>
> > Using [SimUnet], researchers can test how data visualisations and elements of organ offer data impact decision-making when all other details of the organ offer and candidate remain the same.
>
> `continued in next comment...`

---

> > ### Author Response · Authors · 2022-11-10
> > **Thank you for your review (part 2)**
> >
> > Which is clearly a different goal than AllSim. However, they _can_ interact! Currently, AllSim is set to accept each offer that the tested policy proposes. The `policy.select` function allows calls to external systems-- such as SimUnet --to add offer-acceptance decisions into the evaluation strategy should the practitioner so desire (a point can be made about whether or not this would add unwanted noise to the evaluation process). The pseudo-code for such an inclusion would look like:
> >
> > ```python
> > def _propose_match(self, organs: np.ndarray, recipients: np.ndarray) -> Bool:
> >     return simunet.offer(organs, recipients)
> >
> > def select(self, organs: np.ndarray) -> Tuple[np.ndarray, np.ndarray]:
> >     # 1. create datastructure that maps organs to
> >     #    patients to accept status, and initialise
> >     #    accept status to False.
> >     organs_to_patients = ...
> >
> >     # 2. create not proposed list and initialise
> >     #    on self.waitlist
> >     not_proposed = self.waitlist
> >
> >     # 3. iterate through organs_to_patients until
> >     #    all organs have either an accepted status
> >     #    or are rejected by all patients on self._intermediate_waitlist
> >     left_over_organs = organs
> >     while False is in organs_to_patients["decisions"] and len(not_proposed) > 0:
> >         organs, recipients = self._apply_policy_logic(left_over_organs, not_proposed)
> >         accepted = self._propose_match(organs, recipients)
> >
> >         left_over_organs = left_over_organs[np.invert(accepted)]
> >         not_proposed = left_over_recipients[np.invert(accepted)]
> >
> >         organs_to_patients.add((organs[accepted], recipients[accepted], np.repeat(True, len(recipients))))
> >
> >
> >     # 4. perform specific policy logic to maintain self.waitlist
> >     ...
> >
> >     return organs_to_patients["organs"], organs_to_patients["recipients"]
> >
> >
> > ```
> >
> >
> > __*Changes to our manuscript:*__
> > * We included the many references which we cite above, illustrating the size of the research gap we aim to fill.
> > * We discuss SimUnet in detail in our paper as well as provide above suggestion in our appendix F
> >
> >
> >
> > ## (B) Generalisation of AllSim
> >
> > In this part of our rebuttal, we respond to the following comment you make:
> >
> > > Does the simulator generalize to different settings of organ matching with simple tunable knobs?
> >
> > **Short answer: Yes.**
> >
> > **Longer answer:**
> >
> > We give two different examples: (i) vaccine distribution during the COVID-19 pandemic; and (ii) clinician's time in the ICU during the pandemic.
> >
> > ### (i) COVID-19
> >
> > Consider the vaccine distribution setting during the COVID-19 pandemic, which is in many aspects very different from organ allocation:
> >
> > * __[uniqueness]__ Vaccines are not unique
> > * __[batch arrival]__ Vaccines' availability comes in waves (they are ordered at a country level and arrive in batches)
> > * __[changing environment]__ While the vaccine distribution policies are quite simple (e.g. elderly first), the environment (a new virus variant may incur a different response) is changing rapidly
> >
> > AllSim provides many levels of abstraction, allowing all of the above unique properties of vaccine distribution. AllSim is perfectly capable of modelling this scenario given the following:
> >
> > * __[uniqueness]__ As all vaccines are the same, we no longer require a density of resources as we required for organ allocation. This can be done by implementing a dummy-density that always returns 1 (or the type of vaccine).
> > * __[batch arrival]__ Batch arrival simply requires a multiplier. For example, if the Poisson process samples a value of 2 on one day, we could simply interpret this as two batches of 1000 doses.
> > * __[changing environment]__ AllSim is built for this! One could hot-swap (or gradually change) the counterfactual model: a first that is trained on data collected during variant A, and another during variant B. Changing the counterfactual model from A to B would simulate a new virus variant emerging.
> >
> > These implementation details are relatively simple to implement and easily done using AllSim’s modular API.
> >
> > ### (ii) ICU arrival
> >
> > While not necessarily a problem in vaccine distribution, recipient arrival in the ICU in a setting of infectious disease (such as COVID-19), is definitely different as compared to the organ-allocation setting. With organ-allocation, we can safely assume a Poisson process for recipient arrival as recipients enter the system independently. This is of course not true in an infectious disease scenario: one recipient arriving may indicate higher infection rate. As such, recipients _do not_ arrive independently.
> >
> > `continued in next comment...`

---

> > > ### Author Response · Authors · 2022-11-10
> > > **Thank you for your review (part 3)**
> > >
> > > With the above, it is clear that we can no longer rely on a Poisson arrival process for recipients entering the system. Instead, to accurately model a situation of infectious disease, we recommend using a Hawkes process. To further illustrate, we include some code below showing exactly how one may go about including such a Hawkes process in AllSim.
> > >
> > > ```
> > > class HawkesProcess(PoissonProcess):
> > >         def __init__(
> > >                 self,
> > >                 lam: float=.1,
> > >                 update_lam: Callable[[int], float]=lambda t: t,
> > >                 delta: float=.1,
> > >                 a: float=.2
> > >             ):
> > >
> > >             assert a >= 0, "a should be larger than or equal to 0"
> > >             assert delta > 0, "delta should be larger than 0"
> > >
> > >             super().__init__(lam, update_lam)
> > >
> > >             self.a, self.delta = a, delta
> > >             self._samples = []
> > >
> > >         def get_lam_unnormalized(self, t: int) -> float:
> > >             return self._baseline_lam + np.sum(
> > >                 self.a * self.beta * np.exp(-beta * (t - self._samples[self._samples < t])))
> > >
> > >         def progress(self, t: int, neu: float=1) -> int:
> > >             self.lam = neu * self.get_lam_unnormalized(t) # eqs. (5, 6)
> > >             sample = np.random.poisson(lam=self.lam)
> > >             self._samples.append(sample)
> > >             return sample
> > >
> > > ```
> > >
> > > Naturally, if recipient arrival is indeed dependent on previous arrivals, simply learning the arrival process (as we have for figure 3) should model such a self-exciting process automatically. Furthermore, the above implementation is a simple linear univariate Hawkes process. We refer to `hawkeslib` [(https://github. com/canerturkmen/hawkeslib)](https://github. com/canerturkmen/hawkeslib) for implementations of more types of Hawkes processes.
> > >
> > > **_The above two scenarios vary greatly from our running organ allocation example, yet are easily implemented in AllSim. For more information regarding these settings, we refer to Appendix~E._**
> > >
> > > ## (C) Simulation parameters and adoption
> > >
> > > We now turn to the final comment which (we believe) mainly concerns adoption of AllSim in the clinical community (emphasis is ours):
> > >
> > > > [...] tuning these knobs could be challenging for the practitioners as the outcome might depend on a variety of indicators for individual patients. I think it would be great if the authors can perform a _human evaluation study_ and report statistics in terms of how much value the actual practitioners see in this simulator. If it does not pass the bar of practitioner’s expectation, then the counterfactual evaluation in the simulator might not be of practical use. On the other hand, I was expecting some _guidance_ on how to tune the knobs of the simulator, especially for accelerating research progress in this domain.
> > >
> > >
> > > We will split our response into three parts for clarity:
> > >
> > > - (C.1) Human evaluation study
> > >
> > > - (C.2) Guidance for using AllSim – _"Getting started" guide_
> > >
> > > - (C.3) Guidance for using AllSim – _Documentation_
> > >
> > > ### (C.1) Human evaluation study.
> > >
> > > The purpose of a human evaluation study would be (to use the reviewer's own words): _"[to gather] how much value the actual practitioners see in this simulator". _We agree with this!_ However, we wish to clarify that AllSim's conception is actually _due to_ criticism from the clinical community on current practices. We have gathered these criticisms with many conversations we had with many more clinicians (both practising and researching), spread over multiple countries and continents.
> > >
> > > That is to say that these criticisms have directly led to many of the properties and features of AllSim, and there has been continual human evaluation and discussion throughout its development with clinical partners.
> > >
> > > Naturally, due to ICLR's anonymous submission policy, we are not able to acknowledge these clinicians at this stage, but will upon acceptance. However, perhaps the reviewer is interested in published literature that acknowledges the problems AllSim aims to resolve. From Gotlieb et al. (2022) which criticises evaluation in donor-organ allocation:
> > >
> > > > [...] given the additional complexities introduced by ML algorithms, it should be ensured that models undergo rigorous but fair external validation in cohorts or simulated data.
> > >
> > > Warranting interested in specific evaluation strategies in this domain.
> > >
> > > `continued in next part...`

---

> > > > ### Author Response · Authors · 2022-11-10
> > > > **Thank you for your review (part 4)**
> > > >
> > > > From the same work, we also see that the clinical community is interested in seeing how environment changes impact policy effect:
> > > >
> > > > > For more equitable systems, studies should also consider non-clinical variables of transplantation like geographic disparity, physical compatibility between donor and recipients, and resource availability, all of which may significantly impact transplant outcomes.
> > > >
> > > > In particular, we address the above comment with:
> > > >
> > > > * __[geographic disparity.]__ As a feature of both patient and donor, we can directly interact with the density of each object directly (using `allsim.infer.Density`). By changing the density (or the conditional density through `allsim.infer.Condition`), we can immediately see the effect of different disparities influencing the tested policy. Furthermore, AllSim even allows a separate arrival process per condition through the `allsim.infer.System` API.
> > > >
> > > > * __[physical compatibility.]__ This is directly modelled by the counterfactual outcome model. Essentially, the outcome model is a function of exactly the compatibility between resource and recipient where compatibility is expressed in terms of reception outcome.
> > > >
> > > > * __[resource availability.]__ Finally, resource availability is modelled using our arrival processes, using `allsim.infer.PoissonProcess}` (or any other type of arrival process, discussed above in this response).
> > > >
> > > > We stress that above criticism is not ours; it is _"gathered from the actual practitioners"_. AllSim is our answer to the need expressed by the clinical community.
> > > >
> > > > ### (C.2) "getting started" guide
> > > >
> > > >  To use AllSim, a user requires at least a dataset of the following type: $\mathcal{D} \coloneqq \{(X_t, R_u, Y) : t, u \in \mathbb{N}_+\}$, with $t, u$ indicating the time of arrival. In principle, this is sufficient to start using AllSim already. In fact, this is exactly what we provided AllSim in our experiment in Figure 4.  Let us elaborate:
> > > >
> > > > AllSim requires to specify the following components:
> > > >
> > > > * A counterfactual model
> > > > * The patient and resource arrival processes
> > > > * The patient and resource densities
> > > >
> > > > Components 1 and 3 are easily implemented using existing packages like `econml` [(https://econml.azurewebsites.net)](https://econml.azurewebsites.net) and  `scikit-learn` [(https://scikit-learn.org/stable/)](https://scikit-learn.org/stable/), respectively. They only require to use the `model.fit` API to be applied to the users' data:
> > > >
> > > > ```
> > > >     counterf = econml.BaseCateEstimator()
> > > >     counterf.fit(Y, R, X)
> > > >
> > > >
> > > >     density_X, density_R = allsim.infer.KDEDensity(), allsim.infer.KDEDensity()
> > > >     density_X.fit(X)
> > > >     density_R.fit(R)
> > > > ```
> > > >
> > > > The only place where we need AllSim specifically is when we create the arrival processes, and the eventual simulation. Creating an arrival process on data, first requires us to regress the arrival probability on time:
> > > >
> > > > ```
> > > >     from sklearn.preprocessing import PolynomialFeatures
> > > >     from sklearn.linear_model import LinearRegression
> > > >     from sklearn.pipeline import Pipeline
> > > >
> > > >     func_X = function(Pipeline([
> > > >         ('poly', PolynomialFeatures(degree=6)),
> > > >         ('linear', LinearRegression(fit_intercept=True))]))
> > > >
> > > >     # with amount() a function that returns how many of X arrived at t in D
> > > >     func_X.fit(t, amount(X, t))
> > > >
> > > >     # with the arrival processes
> > > >     patient_process = allsim.infer.PoissonProcess(func_X)
> > > >     resource_process = allsim.infer.PoissonProcess(func_R)
> > > >
> > > >     patient_system = allsim.infer.PoissonSystem(density_X, patient_process)
> > > >     resource_system = allsim.infer.PoissonSystem(density_R, resource_process)
> > > > ```
> > > >
> > > > The simulation is then created as:
> > > > ```
> > > >     simulation = asim.sim.Sim(
> > > >         resource_system=resource_system,
> > > >         patient_system=patient_system,
> > > >         inference=counterf
> > > >     )
> > > > ```
> > > >
> > > > The only thing left is to actually run the policy against our created simulation:
> > > >
> > > > ```
> > > >     df = simulation.simulate(policy, T=100) # thats it!
> > > > ```
> > > >
> > > > `continued in next comment...`

---

> > > > > ### Author Response · Authors · 2022-11-10
> > > > > **Thank you for your review (part 5)**
> > > > >
> > > > > ### (C.3) Documentation
> > > > >
> > > > > The above example gives a simple "getting started" example of AllSim. Of course, the true power of AllSim is the ability to change the environment to a user's preference and using counterfactual inference to provide a policy's outcome had it been applied in the specified environment. While we believe AllSim is developed in a user-friendly way, doing this is more involved that what is shown above. As such, an important element to further increase adoption as a thorough and clear documentation. This documentation will be provided as part of open-sourcing AllSim, as is common practice in many open-source projects.
> > > > >
> > > > > While we hope the above will clear things up on how one could get easily started with AllSim, we do want to reiterate that there will also be issue tracking on the future open-source GitHub page (publicised upon acceptance), and constant improvement.
> > > > >
> > > > > **_Changes to our manuscript_**:
> > > > > We have now included our ``getting started'' guide as a subsection to ``Using AllSim'' in our appendix C. Much more detail on using AllSim will naturally be provided through detailed documentation, as is (or should be) the standard in open-source projects.
> > > > >
> > > > > **_We hope we have clarified the comments you made in your rebuttal. Should there still be remaining questions, please do not hesitate to ask!_**
> > > > >
> > > > > ### Additional references
> > > > >
> > > > > W Ray Kim, Ajitha Mannalithara, Julie K Heimbach, Patrick S Kamath, Sumeet K Asrani, Scott W Biggins, Nicholas L Wood, Sommer E Gentry, and Allison J Kwong. Meld 3.0: the model for end-stage liver disease updated for the modern era. Gastroenterology, 161(6):1887–1895, 2021
> > > > >
> > > > > Osvald Nitski, Amirhossein Azhie, Fakhar Ali Qazi-Arisar, Xueqi Wang, Shihao Ma, Leslie Lilly, Kymberly D Watt, Josh Levitsky, Sumeet K Asrani, Douglas S Lee, et al. Long-term mortality risk stratification of liver transplant recipients: real-time application of deep learning algorithms on longitudinal data. The Lancet Digital Health, 3(5):e295–e305, 2021
> > > > >
> > > > > Uri Kartoun. Towards optimally replacing the current version of meld. Journal of Hepatology, 2022 Ina Jochmans, Marieke van Rosmalen, Jacques Pirenne, and Undine Samuel. Adult liver allocation in eurotransplant. Transplantation, 101(7):1542–1550, 2017
> > > > >
> > > > > David Goldberg, Alejandro Mantero, Craig Newcomb, Cindy Delgado, Kimberly Forde, David Kaplan, Binu John, Nadine Nuchovich, Barbara Dominguez, Ezekiel Emanuel, et al. Development and validation of a model to predict long-term survival after liver transplantation. Liver transplantation, 27(6):797–807, 2021
> > > > >
> > > > > James Neuberger, Alex Gimson, Mervyn Davies, Murat Akyol, John O’Grady, Andrew Burroughs, Mark Hudson, UK Blood, et al. Selection of patients for liver transplantation and allocation of donated livers in the uk. Gut, 57(2):252–257, 2008
> > > > >
> > > > > Jeroen Berrevoets, James Jordon, Ioana Bica, Alexander Gimson, and Mihaela van der Schaar. OrganITE: Optimal transplant donor organ offering using an individual treatment effect. In Advances in Neural Information Processing Systems, volume 33, pp. 20037–20050. Curran Associates, Inc., 2020
> > > > >
> > > > > Jeroen Berrevoets, Ahmed Alaa, Zhaozhi Qian, James Jordon, Alexander ES Gimson, and Mihaela Van Der Schaar. Learning queueing policies for organ transplantation allocation using interpretable counterfactual survival analysis. In International Conference on Machine Learning, pp. 792–802. PMLR, 2021
> > > > >
> > > > > Jinsung Yoon, Ahmed Alaa, Martin Cadeiras, and Mihaela Van Der Schaar. Personalized donor-recipient matching for organ transplantation. In Proceedings of the AAAI Conference on Artificial Intelligence, volume 31, 2017
> > > > >
> > > > > Katie L Connor, Eoin D O’Sullivan, Lorna P Marson, Stephen J Wigmore, and Ewen M Harrison. The future role of machine learning in clinical transplantation. Transplantation, 105(4):723–735, 2021
> > > > >
> > > > > Dennis Medved, Pierre Nugues, and Johan Nilsson. Simulating the outcome of heart allocation policies using deep neural networks. In 2018 40th Annual International Conference of the IEEE Engineering in Medicine and Biology Society (EMBC), pp. 6141–6144. IEEE, 2018a

---

> > > > > > ### Author Response · Authors · 2022-11-10
> > > > > > **references continued**
> > > > > >
> > > > > > Manuel Dorado-Moreno, María Pérez-Ortiz, Pedro A Gutiérrez, Rubén Ciria, Javier Briceño, and César Hervás-Martínez. Dynamically weighted evolutionary ordinal neural network for solving an imbalanced liver transplantation problem. Artificial Intelligence in Medicine, 77:1–11, 2017
> > > > > >
> > > > > > Dennis Medved, Mattias Ohlsson, Peter Höglund, Bodil Andersson, Pierre Nugues, and Johan Nilsson. Improving prediction of heart transplantation outcome using deep learning techniques. Scientific reports, 8 (1):1–9, 2018b
> > > > > >
> > > > > > Kyung Don Yoo, Junhyug Noh, Hajeong Lee, Dong Ki Kim, Chun Soo Lim, Young Hoon Kim, Jung Pyo Lee, Gunhee Kim, and Yon Su Kim. A machine learning approach using survival statistics to predict graft survival in kidney transplant recipients: a multicenter cohort study. Scientific reports, 7(1):1–12, 2017
> > > > > >
> > > > > > Dimitris Bertsimas, Vivek F Farias, and Nikolaos Trichakis. Fairness, efficiency, and flexibility in organ allocation for kidney transplantation. Operations Research, 61(1):73–87, 2013
> > > > > >
> > > > > > Dimitris Bertsimas, Vivek F Farias, and Nikolaos Trichakis. On the efficiency-fairness trade-off. Management Science, 58(12):2234–2250, 2012
> > > > > >
> > > > > > Amir Elalouf, Yael Perlman, and Uri Yechiali. A double-ended queueing model for dynamic allocation of live organs based on a best-fit criterion. Applied Mathematical Modelling, 60:179–191, 2018
> > > > > >
> > > > > > John P Dickerson, Ariel D Procaccia, and Tuomas Sandholm. Failure-aware kidney exchange. In Proceedings of the fourteenth ACM conference on Electronic commerce, pp. 323–340, 2013
> > > > > >
> > > > > > Mustafa Akan, Oguzhan Alagoz, Baris Ata, Fatih Safa Erenay, and Adnan Said. A broader view of designing the liver allocation system. Operations research, 60(4):757–770, 2012
> > > > > >
> > > > > > Patrick S Kamath, Russell H Wiesner, Michael Malinchoc, Walter Kremers, Terry M Therneau, Catherine L Kosberg, Gennaro D’Amico, E Rolland Dickson, and W Ray Kim. A model to predict survival in patients with end-stage liver disease. Hepatology, 33(2):464–470, 2001
> > > > > >
> > > > > > Andres E Ruf, Walter K Kremers, Lila L Chavez, Valeria I Descalzi, Luis G Podesta, and Federico G Villamil. Addition of serum sodium into the meld score predicts waiting list mortality better than meld alone. Liver Transplantation, 11(3):336–343, 2005
> > > > > >
> > > > > > Neta Gotlieb, Amirhossein Azhie, Divya Sharma, Ashley Spann, Nan-Ji Suo, Jason Tran, Ani Orchanian-Cheff, Bo Wang, Anna Goldenberg, Michael Chassé, et al. The promise of machine learning applications in solid organ transplantation. NPJ digital medicine, 5(1):1–13, 2022

---

> ### Comment · Area_Chair_Ruz4 · 2022-11-21
> **Any comments to the responses from authors?**
>
> Dear Reviewer uUjW,
>
> Thank you very much for your review.  The authors have provided very detailed responses to your concerns.  How did they change your evaluation, particularly on the novelty and practical adoption?

---

> > ### Comment · Reviewer_uUjW · 2022-11-22
> > **Scores unchanged**
> >
> > Dear Authors,
> >
> > Thank you very much for the detailed response. The responses clarified majority of my questions. However, given that it proposes a purely simulation model (without any theoretical results/properties and experimental on real-world testbed), I think a bare minimum requirement is to show that the simulator is ready for adoption by a significant proportion of practitioners (obviously a medium scale human evaluation is required for that). Until we have upvote from practitioners, it is hard to get confidence that research community will fully align and start using this simulator. For instance, it is not clear to me at this point that I can fully trust this simulator for counter-factual evaluation to emulate real-world evolution. Therefore, similar to other reviewers, I am also a little unsure about the novelties for ICLR standard -- so, I would like to keep my scores unchanged.

---

### Official Review · Reviewer_ACC7 · 2022-10-26

**Confidence:** 5
**Correctness:** 1
**Technical Novelty And Significance:** 1
**Empirical Novelty And Significance:** 1
**Recommendation:** 1

**Clarity, Quality, Novelty And Reproducibility:**

Although the quality of presentation early in the paper is good. The latter parts of the paper become much weaker. Presenting code fragments is not the right level for academic conferences.

The idea is not novel and has been done extensively in the literature.

The authors use their own notation in places with no explanation of what it is or how it works.

**Strength And Weaknesses:**

Strengths:
 - The English in the paper is clear and well structured
 - The ideas are clearly presented
Weaknesses:
 - The work presents ideas which have been covered before using formal methods using tools such as PEPA, CARMA AND SPN. Comparison to these tools would seem essential.
 - The results section is only a small number of simple examples and does not compare the work to other approaches.
 - There are many statements in the work which are incorrect. And many which need supporting evidence such as citation to other works.

**Summary Of The Paper:**

The authors present a simple simulation model which they argue can "Learn" from the environment. However, the authors seem unaware of teen area of formal methods and the tools in this area such as PEPA, CARMA OR SPN which this work appears to be a re-invention of.

**Summary Of The Review:**

The paper contains only a simulation approach which has been widely covered in prior work. There is nothing in this work which is novel in the way of ICLR. The authors need to reflect on tools such as PEPA, CARMA and SPN. A more analytical analysis of the solution is needed.

---

> ### Author Response · Authors · 2022-11-10
> **Thank you for your review (part 1)**
>
> _Dear reviewer ACC7, thank you for your review of our work. We have addressed your comments below. For clarity, we have split them into two main themes:_
>
> * (A) Related work
>
> * (B) Using code
>
> ## (A) Related work
>
> You suggest in your review:
>
> > The work presents ideas which have been covered before using formal methods using tools such as PEPA, CARMA AND SPN. Comparison to these tools would seem essential.
>
> Let us discuss each simulator you suggest above. We will focus mainly on providing evidence that AllSim is different from each of the simulators you propose. Please note that, without a formal reference provided, we will assume the simulators we found using the acronym presented by the reviewer are the correct ones.
>
> ### (A.1) Performance Evaluation Process Algebra (PEPA) [[link to sim]](https://www.dcs.ed.ac.uk/pepa/)
>
> The PEPA "about" page highlights that PEPA is a language to model distributed _computer_ systems:
>
> > Jane Hillston's Performance Evaluation Process Algebra (PEPA) is an expressive formal language for modelling distributed systems.
>
> The main feature they propose is a Java Eclipse (IDE) plugin to model "Markovian analysis, continuous-space analysis and simulation". The main purpose then is to use linear algebra to assign computers in the distributed computing setting– a goal completely different from AllSim's. This is further illustrated in their proposed use-cases according to the "about" page:
>
> > The application areas for this work span the subject areas of Informatics and engineering.
> > * Active badge systems (Clark, Gilmore and Hillston, Edinburgh)
> > * Cellular telephone networks (Kloul, Versailles)
> > * Database systems (The STEADY group, Heriot-Watt and Clark, Edinburgh)
> > * Diagnostic expert systems (Ribaudo, Turin)
> > * Elevator systems (Kwiatkowska, Birmingham)
> > * Multimedia traffic characteristics (Bowman, UKC)
> > *  Multiprocessor access-contention protocols (Gilmore, Hillston and Ribaudo)
> > *  Multi-server, multi-queue systems (Harrison, Imperial and Hillston, Edinburgh)
> > *  On-line auction systems (Hillston, Edinburgh and Kloul, Versailles)
> > *  Protocols for fault-tolerant systems (Clark, Gilmore, Hillston and Ribaudo)
> > *  Robotic workcells (Holton, Bradford and Gilmore and Hillston, Edinburgh)
> > *  Software architectures (Pooley, Heriot-Watt and Thomas and Bradley, Durham)
> > *  System-on-chip technology (Eder, Bristol)
>
> *__How is AllSim different?__* While we appreciate the reviewer's suggestion, we hope the above makes it absolutely clear that PEPA is not related to AllSim in any stronger sense than it being designed to simulate a process. Let us summarise the key reasons below:
>
> * __[Different Goal]__ AllSim evaluates an allocation policy in the medical domain; PEPA evaluates the efficiency of a distributed computer system. These goals are not the same.
>
> * __[No data]__  PEPA is a _language_, at no point does PEPA leverage any data to learn an environment to evaluate policies. However, leveraging data for this exact purpose is one of AllSim's main selling points.
>
> * __[No unique resources]__ While we understand that distributed system optimisation shares the objective of limited resource allocation, we believe it is important to reiterate that PEPA does not concern _unique_ resources. The resources PEPA deals with are known a priori and are part of the optimisation objective, completely opposite from AllSim where resources arrive sequentially and uniquely, for the policy to assign in the moment.
>
> * __[Java]__ Perhaps a different programming language may seem like a small thing, however considering that– given a few exceptions such as R-lang or Julia –the ML community uses almost exclusively Python, it is simply not possible to easily incorporate PEPA with existing policy implementations into a practical workflow. Furthermore, even the medical clinical community is moving towards python for this exact reason!
>
> * __[Continues time]__  PEPA evaluates in continuous time, whereas AllSim in discrete time. As resources do not arrive constantly, the only way to model medical scarce resource allocation is in discrete time.
>
> ### (A.2) CARMA Simulation [[link to sim]](https://usdot-carma.atlassian.net/wiki/spaces/CRMSIM/overview)
>
> CARMA, a product from the US Dept. of Transportation, is a simulator to evaluate cooperative driving automation (CDA). From the CARMA "about" page:
>
> > CARMA focuses on improving the transportation system by using emerging automated driving and vehicle-to-everything (V2X) technologies to enhance safety, operational efficiency, and sustainability in moving people and goods.
>
> `continued in next comment...`

---

> > ### Author Response · Authors · 2022-11-10
> > **Thank you for your review (part 2)**
> >
> > They go on to further explain some of the goals of CARMA:
> >
> > > * Establish everything-in-the-loop (XiL) capabilities to support CDA evaluation in a simulation environment.
> > > * Begin building XiL capabilities through open source software through collaboration with the Department of Energy and CARMA community to effectively design and build tools to advance the understanding of CDA’s impact on the transportation system.
> >
> > From the above direct quotes, it should be clear that CARMA is a purpose built simulation specifically for driving automation. In fact their online YouTube video seems to back this up: [link to video](https://www.youtube.com/watch?v=H-7js__UrgQ).
> >
> > *__How is AllSim different?__* We fail to see the relevance of CARMA to our use case. CARMA is purpose built for automated driving. While we understand that cars may be a limited resource, we stress that not _all_ limited resource problems are the same. In our case, we focus on the medical setting where the resources have more properties than simply being unavailable or available. Furthermore, we cannot simply match any resource with any recipient.
> >
> > ### (A.3) Simulated Patient Network (SPN) [[link to sim]](https://www.simulatedpatientnetwork.org)
> >
> > While we were unfamiliar with the previous simulators the reviewer pointed us to, we are actually quite familiar with the SPN. The SPN is part of some medical education programs, and from their web page:
> >
> > > The Simulated Patient Network – SPN (previously, Victorian Simulated Patient Network – VSPN) was established to create a network of individuals with a special interest in simulated patient (SP) methodology. The SPN offers online modules on various aspects of working with SPs involved in health and social care professional education. The modules are likely to be of interest to simulation educators, technicians, SPs and to some extent learners involved in undergraduate and postgraduate training in all health professional disciplines.
> >
> > With simulated patients (SPs), the trainee can interact with an individual (the SP) to gain experience into what it is like to directly interact with a patient:
> >
> > > SPs are individuals trained to portray real patients. SPs contribute to health professional education in many ways. There is a spectrum of SP practices that are expanding. The SPN aims to connect SPs, educators, students, clinicians and others through this website.
> >
> > *__How is AllSim different?__* While technically a simulation (cfr. Wikipedia's definition: "A simulation is the imitation of the operation of a real-world process or system over time"), it should be clear that SPN is not the type of simulation we propose. Our simulation is a computer program, SPN is essentially a role-playing program (for an example interaction, we refer to this YouTube video: [link to video](https://www.youtube.com/watch?v=8khndq0yW64)). We hope that this difference is clear for the reviewer.
> >
> > _We hope this response clarifies that the simulators the reviewer proposes are simply not related in any significant sense. In particular, we hope this highlights that they **do not warrant comparison** in such a space-constrained conference paper as they are designed for different problems, produce different outputs, and work in fundamentally different ways._
> >
> > `continued in next comment...`

---

> > > ### Author Response · Authors · 2022-11-10
> > > **Thank you for your review (part 3)**
> > >
> > > ## (B) Using code
> > >
> > > The reviewer suggests that displaying code-segments in our paper is inappropriate.
> > >
> > > There are two main reasons that we believe that this is a misguided opinion: (i) we propose an open-source framework which is essentially a code-base, as this framework is the topic of our paper, it is only natural that our paper contains code; and (ii) our work is hardly unique in this respect, in fact we find many accepted papers at top conferences, including ICLR, that present code in a similar way.
> > >
> > > __(i) We present a framework.__ While rebutting to other reviewers' comments, it is clear that we should have included many more code fragments. The reason is simple: code describes what words cannot. In our efforts of presenting AllSim--- an open-source simulator written in Python ---it is instructive to include the specific interaction with our simulator that yields certain results, setups, environments, etc. as it later helps readers and users of our paper to reproduce and customise the AllSim environment to their needs. _What else is the purpose of a paper if not to provide clarity about a project?_ In our case, most clarity is achieved by discussing parts of our code.
> > >
> > > __(ii) We are not the first to discuss code.__ Here are a few recent machine learning papers that include code-segments, some of them with citation counts over 20 thousand: : Paszke et al. (2019); Millman & Aivazis (2011); Meurer et al. (2017); Ari & Ustazhanov (2014); Virtanen et al. (2020); Abel (2019); Müller & Behnke (2014); LeDell & Poirier (2020); Abadi et al. (2016); Townsend et al. (2016); Andreux et al. (2020); Jarrett et al. (2021). Please note that while we cited only a few across different publication venues, many more examples are easily found.
> > >
> > > **_We hope that our response has helped to clarify things. However, should there still remain questions, please do not hesitate to ask for further clarification!_**
> > >
> > > ## Additional references
> > >
> > > Adam Paszke, Sam Gross, Francisco Massa, Adam Lerer, James Bradbury, Gregory Chanan, Trevor Killeen, Zeming Lin, Natalia Gimelshein, Luca Antiga, et al. Pytorch: An imperative style, high-performance deep learning library. Advances in neural information processing systems, 32, 2019
> > >
> > > K Jarrod Millman and Michael Aivazis. Python for scientists and engineers. Computing in Science & Engineering, 13(2):9–12, 2011
> > >
> > > Aaron Meurer, Christopher P Smith, Mateusz Paprocki, Ondˇrej ˇCertík, Sergey B Kirpichev, Matthew Rocklin, AMiT Kumar, Sergiu Ivanov, Jason K Moore, Sartaj Singh, et al. Sympy: symbolic computing in python. PeerJ Computer Science, 3:e103, 2017
> > >
> > > Niyazi Ari and Makhamadsulton Ustazhanov. Matplotlib in python. In 2014 11th International Conference on Electronics, Computer and Computation (ICECCO), pp. 1–6. IEEE, 2014
> > >
> > > Pauli Virtanen, Ralf Gommers, Travis E Oliphant, Matt Haberland, Tyler Reddy, David Cournapeau, Evgeni Burovski, Pearu Peterson, Warren Weckesser, Jonathan Bright, et al. Scipy 1.0: fundamental algorithms for scientific computing in python. Nature methods, 17(3):261–272, 2020
> > >
> > > David Abel. simple_rl: Reproducible reinforcement learning in python. In RML@ ICLR, 2019
> > >
> > > Andreas C Müller and Sven Behnke. Pystruct: learning structured prediction in python. J. Mach. Learn. Res., 15(1):2055–2060, 2014
> > >
> > > Erin LeDell and Sebastien Poirier. H2o automl: Scalable automatic machine learning. In Proceedings of the AutoML Workshop at ICML, volume 2020, 2020
> > >
> > > Martín Abadi, Paul Barham, Jianmin Chen, Zhifeng Chen, Andy Davis, Jeffrey Dean, Matthieu Devin, Sanjay Ghemawat, Geoffrey Irving, Michael Isard, et al. TensorFlow: a system for Large-Scale machine learning. In 12th USENIX symposium on operating systems design and implementation (OSDI 16), pp. 265–283, 2016
> > >
> > > James Townsend, Niklas Koep, and Sebastian Weichwald. Pymanopt: A python toolbox for optimization on manifolds using automatic differentiation. Journal of Machine Learning Research, 17(137):1–5, 2016. URL http://jmlr.org/papers/v17/16-177.html
> > >
> > > Mathieu Andreux, Tomás Angles, Georgios Exarchakis, Roberto Leonarduzzi, Gaspar Rochette, Louis Thiry, John Zarka, Stéphane Mallat, Joakim Andén, Eugene Belilovsky, et al. Kymatio: Scattering transforms in python. J. Mach. Learn. Res., 21(60):1–6, 2020
> > >
> > > Daniel Jarrett, Jinsung Yoon, Ioana Bica, Zhaozhi Qian, Ari Ercole, and Mihaela van der Schaar. Clairvoyance: A pipeline toolkit for medical time series. In International Conference on Learning Representations, 2021. URL https://openreview.net/forum?id=xnC8YwKUE3k

---

> ### Comment · Area_Chair_Ruz4 · 2022-11-21
> **Any comments to the responses from authors?**
>
> Dear Reviewer ACC7,
>
> Thank you very much for your review.  The authors have provided very detailed responses to your concerns.  Please read them and let us know how your evaluation changes.  In particular, the authors explain how the proposed approach is different from the existing approaches that you suggested.  Do you now agree on the novelty of the proposed approach?

---

### Decision · Program_Chairs · 2023-01-20

**Decision:**

Reject

**Justification For Why Not Higher Score:**

The novelty of the proposed simulation method is unclear, and the advantages of the proposed method over existing methods are not demonstrated.

**Justification For Why Not Lower Score:**

N/A

**Metareview: Summary, Strengths And Weaknesses:**

This paper contributes a method for running simulations.  The specific problem addressed of allocating transplants is quite important, and the fact that the system has been designed with medical practitioners can suggest that the system will be adopted and that it satisfies the needs for which it has been designed.  This potential impact is a major strength of the paper.

While the proposed simulation module might be novel for a variety of reasons, the advantages of the proposed method over the status quo are not well demonstrated in the paper.  For example, one could take multiple resource-allocation settings, and
- simulate resource arrivals and distributions in a reasonable way, using available datasets. (make sure that these simulations align with observed behaviors)
- simulate resource allocation policies used in practice, and verify that their behavior/outcomes aligns with real-world observations
- simulate new resource allocation policies (e.g., reinforcement learning)
- use a "status quo" simulation method from the literature, and demonstrate why this is inferior to the proposed method

Also, the subject of the paper is not the core topic of ICLR.  While this does not prohibit publications in ICLR, other communities might evaluate the paper differently.